



# LISFLOOD-FP 8.1: New GPU accelerated solvers for faster fluvial/pluvial flood simulations

Mohammad Kazem Sharifian[1,2], Georges Kesserwani[1], Alovya Ahmed Chowdhury[1], Jeffrey Neal[3], and Paul Bates[3]

[1]Department of Civil and Structural Engineering, The University of Sheffield, Western Bank, Sheffield, UK
[2]Risk Management Solutions Ltd., London, UK
[3]School of Geographical Sciences, University of Bristol, Bristol, UK

*Correspondence to*: Mohammad Kazem Sharifian (mohammad.sharifian@rms.com) and Georges Kesserwani (g.kesserwani@sheffield.ac.uk)

**Abstract.** The local inertial two-dimensional (2D) flow model on LISFLOOD-FP, so-called ACC uniform grid solver, has been widely used to support fast, computationally efficient fluvial/pluvial flood simulations. This paper describes new releases, on LISFLOOD-FP 8.1, for parallelised flood simulations on the graphical processing units (GPU) to boost efficiency of the existing parallelised ACC solver on the central processing units (CPU) and enhance it further by enabling a new non-uniform grid version. The non-uniform solver generates its grid using the multiresolution analysis (MRA) of the 15 multiwavelets (MWs) to a Galerkin projection of the digital elevation model (DEM). This sensibly coarsens the resolutions where the local topographic details are below an error threshold $\varepsilon$ and allows to properly adapt classes of land use. Both the grid generator and the adapted ACC solver on the non-uniform grid are implemented in a GPU new codebase, using the indexing of Z-order curves alongside a parallel tree traversal approach. The efficiency performance of the GPU parallelised uniform and non-uniform grid solvers are assessed for five case studies, where the accuracy of the latter is explored for $\varepsilon = 20 \quad 10^{-4}$ and $10^{-3}$ in terms of how close it can reproduce the prediction of the former.

Keywords: flood modelling, non-uniform grid generator, GPU implementation, catchment and urban scales, practical recommendations.

## 1 Introduction

LISFLOOD-FP is a raster-based open-source hydrodynamic modelling framework that has been applied in many fields of earth sciences, including morphodynamic modelling (Coulthard et al., 2013; Ziliani et al., 2020), urban drainage modelling (Wu et al., 2018; Yang et al., 2022), population mapping (Zhu et al., 2020), coastal flooding (Hirai and Yasuda, 2018; Seenath, 2018; Irwan et al., 2021), uncertainty quantification (Liu and Merwade, 2018; Beevers et al., 2020; Jafarzadegan et al., 2021; Karamouz and Mahani, 2021; Yin et al., 2022; Zeng et al., 2022) and coupled hydrological-hydraulic modelling 30  (Siqueira et al., 2018; Towner et al., 2019; Hoch et al., 2019; Rajib et al., 2020; Makungu and Hughes, 2021; Nandi et al., 2022). It has undergone extensive developments and testing since its conception (e.g., Bates and De Roo, 2000; Hunter et al., 2005; Bates et al., 2010; Sosa et al., 2020; Shustikova et al., 2020; Zhao et al., 2020), becoming a state-of-the-art tool for flood modelling applications at urban, catchment, regional and continental scales (e.g., Amarnath et al., 2015; Chaabani et al., 2018; Zhu et al., 2019; O'Loughlin et al., 2020; Zare et al., 2021; Bessar et al., 2021; Chone et al., 2021; Asinya et al., 35  2021; Zhao et al., 2021).

LISFLOOD-FP includes a variety of numerical schemes to solve the two-dimensional (2D) shallow water equations with different levels of sophistication, ranging from the simplistic diffusive wave scheme to more complex finite volume and Galerkin schemes solving the full shallow water equations (Shaw et al., 2021). The ACC solver is mathematically more complex than the diffusive wave scheme due to considering the acceleration term in the momentum conservation equation 40  (Bates et al., 2010). Its extra complexity pays off with larger and more stable time steps (Hunter et al., 2006; Hunter et al.,



2008; Wang et al., 2011). As opposed to the more complex schemes that solve the full shallow water equations, the ACC solver neglects the advection terms in the momentum conservation equation. The ACC solver is therefore faster to run: it uses a simple finite difference/volume numerical scheme in a staggered approach that evaluates continuous discharges across the grid elements to then achieve an element-wise update of averaged water levels (Bates et al., 2010). This staggered

approach allows for retaining a compact computational stencil allowing to avoid the costs of deploying an approximate Riemann solver. Consequently, the ACC solver has been favoured to run fast and sufficiently accurate simulations for fluvial floods from rivers and/or for rainfall-driven pluvial floods, featured with gradually propagating 2D flows over rough areas with Manning parameters higher than 0.03 s/m$^{1/3}$ (Neal et al., 2012b; de Almeida and Bates, 2013). It has received further developments to sustain its utility and speed for such applications: for instance, the ACC solver has been enabled with a

subgrid channel model to integrate the flow from river channels with any width below that of the grid resolution (Neal et al., 2012a), and its efficiency maximised on the Central Processing Unit (CPU) with shared-memory parallelisation with an optimised treatment of wet-dry zones (Neal et al., 2018). Still, the ACC solver can be further sustained with alternative speed-up measures to enable faster and more efficient simulations across a wide range of spatial scales and grid resolutions (Guo et al., 2021).

One useful speed-up measure is the capability of running the ACC solver on a non-uniform grid with sensibly coarsened resolutions in the smooth portions of the digital elevation model (DEM) alongside proper distribution of different classes of land use such as, for example, different classes of Manning parameters. Such a non-uniform grid can be generated by while introducing one error threshold parameter, $\varepsilon$: by reshaping the DEM into a planar Galerkin projection and then applying the multiresolution analysis (MRA) of multiwavelets (MWs) (Kesserwani et al., 2019; Kesserwani and Sharifian,

2020; Özgen-Xian et al., 2020). The scheme of the ACC solver can readily be adapted to the non-uniform grid, hereafter referred to as the non-uniform grid solver. Another useful speed-up measure is to further exploit the parallelism of the GPU architecture. This can be achieved without any complication for the uniform solver due to the uniformity of the indexing system, using nested grid-stride loops in the existing GPU codebase developed in Shaw et al. (2021). However, an ad-hoc codebase must be designed to make the non-uniform grid solver run efficiently on the GPU, namely, to accommodate the

irregular grid resolution layout featured in the MRA process of the grid generator and handle data reindexing, access, distribution and update on the non-uniform grid.

This paper describes the design and integration of a non-uniform grid solver on LISFLOOD-FP 8.1 that runs on the GPU. The portability of the uniform solver to existing GPU codebase (Shaw et al. 2021) is straightforward and won't therefore be presented. In Sect. 2, the non-uniform grid solver is described, with a focus on: (i) the grid generator that

sensibly coarsens local DEM resolutions, (ii) the adaptation of the ACC scheme to conserve fluxes across non-uniformly sized elements, and (iii) the ad-hoc codebase to efficiently implement both the grid generator and the adapted scheme on the GPU. In Sect. 3, the accuracy of the non-uniform solver is assessed for five realistic flooding case studies for selected $\varepsilon$ values, considering the level of closeness of its predictions to those of the uniform solver. The ratio of runtimes is used to evaluate the speed-up gain from the GPU version of the uniform solver with respect to the CPU version, and of the non-

uniform solver with respect to the uniform solver on the GPU. In Sect. 4, conclusions are drawn on the potential utility of the GPU accelerated local inertial solvers. The codes of these solvers are openly available (https://doi.org/10.5281/zenodo.6912932) for non-commercial use under the GPLv3.0 licence, with a user guide available on www.seamlesswave.com/LISFLOOD8.0.

## 2 Non-uniform solver on the GPU architecture

LISFLOOD-FP 8.1 includes a non-uniform grid solver that runs on the GPU architecture. The solver is made of two components: the grid generator that applies the MRA of MWs to a planar Galerkin projection of DEM data to sensibly





coarsen resolutions and accordingly adapt classes of land use parameters (Sect. 2.1); and the adapted ACC scheme on the generated grid to aggregate the discharges across non-uniformly sized elements of the grid (Sect. 2.2). Both the grid generator and the adapted ACC scheme are packed in a new codebase to efficiently implement the non-uniform grid solver

on the GPU (Sect. 2.3).

### 2.1  Non-uniform grid generator with spatially-varying distribution of friction

Given a 2D rectangular domain spanned by a raster-formatted DEM at resolution R, with $p_x \times p_y$ pixels ($p_x$ and $p_y$ being the number of pixels in the $x$ and $y$ directions, respectively), a maximum refinement level $L$, is deduced such that $2^L \geq p_x$ and $2^L \geq p_y$. Then, a hierarchy of grids of decreasing refinement levels $L$, $L$ - 1, …, 0 is assumed in a dyadic manner, so that the

finest grid has $2^L \times 2^L$ elements at resolution $R$, while the coarsest grid has only a single element at resolution $2^L R$. Figure 1a shows an example of a hierarchy of grids with $L = 3$. The grid generator starts by reshaping the topography data over each element on level $L$ as a planar Galerkin projection (Shaw et al., 2021). The planar data are then recursively produced for the elements on the coarser grids at refinement levels $L$ - 1, $L$ - 2, …, 1, 0 in a process known as "encoding" (Kessewani and Sharifian, 2020). Encoding produces the "details", representing the encoded difference between the topography data across

two subsequent refinement levels, which are generated from the MWs supplements to the planar topographic representation. These details become increasingly significant where there is sharp variation in the topography data, but remain small otherwise. The details are then analysed by comparing their magnitude to a user-specified error threshold $\varepsilon$, to only retain a tree-like structure of elements with significant details. Figure 1b shows such a tree-like structure, with "leaf elements" at the end of each branch, specified in colour depending on their refinement level. These leaf elements are then used to assemble a

non-uniform grid made of non-overlapping elements at various resolutions, as shown in Figure 1c. The grid generator can also be used to adapt a spatially-varying distribution of land use parameters on the assembled grid. Here, it was used to adapt different classes of Manning's friction parameters by applying the MRA to a raster-formatted Manning's data map that is initially available at resolution level $L$.

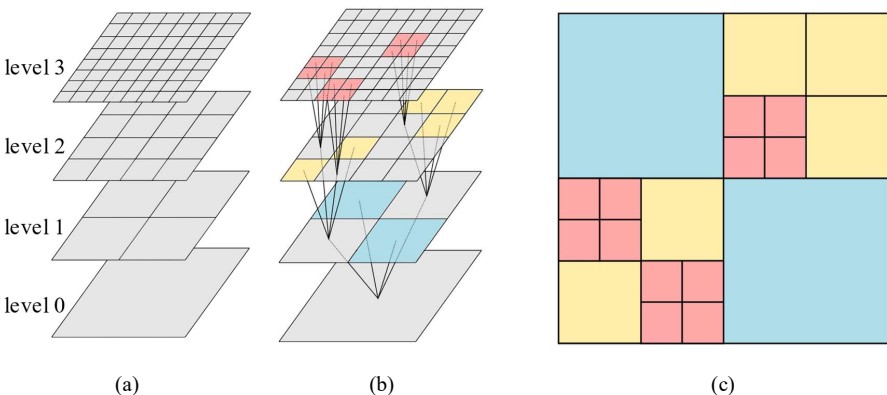

<table>
<tr><td>(a)</td><td>(b)</td><td>(c)</td></tr>
</table>

**Figure 1. Outline of MRA: (a) hierarchy of grids, (b) elements associated with the leaf elements of a tree of significant details and**
**(c) non-uniform grid assembled from these elements.**

### 2.2  Adaptation on the non-uniform grid

The non-uniform grid (e.g. Figure 1c) includes non-uniformly sized faces as each element can have a coarser neighbour or multiple finer neighbours. To adapt the scheme of the ACC solver to such a non-uniform grid, amendments must be made to aggregate the discharge estimates across non-uniformly sized faces from the side of the elements with finer resolution levels.

Figure 2 shows an example of an element at position ($i,j$) (blue element) that is adjacent to an element of one level coarser resolution (yellow element, $e3$) and two elements on two levels finer resolution (red elements, $e1$ and $e2$) over its eastern face. For instance, the $x$-directional discharge, $Q_x$ (m³/s), across element ($i,j$) and element $e1$, is calculated as:



$$Q_{x_{e1}}^{n+1} = \frac{q_{x_{e1}}^{n} - gh_{f\frac{\Delta t}{1/2(\Delta x+\Delta x_{e1})}}\left(\eta_{i,j}^{n}-\eta_{e1}^{n}\right)}{\left(1+g\Delta t n^2|q_{x_{e1}}^{n}|/h_f^{7/3}\right)}\Delta x_{e1} \tag{1}$$

where $\Delta x$ is the size of element $(i,j)$, $\Delta x_{e1}$ is the size of element $e1$, $g$ (m/s$^2$) is the gravity acceleration, $q_{x_{e1}}^{n}$ (m$^2$/s) is the discharge per unit width, $n_M$ (s/m$^{1/3}$) is Manning's friction parameter, $\eta = h + z$ is water surface elevation at each element defined from water depth $h$ (m) and topography $z$ (m) at each element, and $h_f$ (m) is calculated as:

$$h_f = max(h_{i,j} + z_{i,j}, h_{e1} + z_{e1}) - max(z_{i,j}, z_{e1}) \tag{2}$$

After computing the discharges over all other three neighbouring interfaces $e1$, $e2$ and $e3$, the total discharge across the eastern face of the element $(i,j)$ is aggregated as

$$Q_{x_{i+1/2,j}}^{n+1} = Q_{x_{e1}}^{n+1} + Q_{x_{e2}}^{n+1} + Q_{x_{e3}}^{n+1} \tag{3}$$

to be used in mass conservation equation to evolve the water depth as

$$h_{i,j}^{n+1} = h_{i,j}^{n} + \Delta t \frac{Q_{x_{i-1/2,j}}^{n+1} - Q_{x_{i+1/2,j}}^{n+1} + Q_{y_{i,j-1/2}}^{n+1} - Q_{y_{i,j+1/2}}^{n+1}}{A_{i,j}} \tag{4}$$

where $Q_{x_{i-1/2,j}}^{n+1}$, $Q_{y_{i,j-1/2}}^{n+1}$ and $Q_{y_{i,j+1/2}}^{n+1}$ are discharge estimates at the western, southern and northern faces, respectively, computed in the same manner, and $A_{i,j}$ is the area of element $(i,j)$.

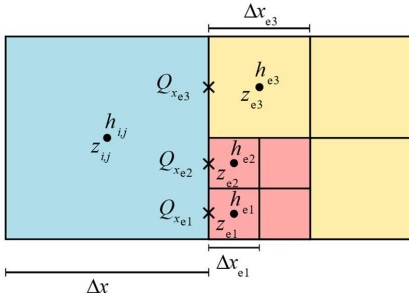

**Figure 2. Schematic layout of neighbouring elements on a non-uniform grid, where a coarse element (in blue) is adjacent to one- and two-level finer elements (in yellow and red, respectively).**

### 2.3 GPU implementation

The grid generator of the non-uniform solver entails an irregular grid resolution layout to perform the recursive operations of the MRA process and to reindex, access, distribute and update the flow data on the non-uniform grid via the adapted scheme of the ACC solver. In what follows, the GPU implementation of the non-uniform solver is presented with a focus on the parallel handling of the MRA process (Sect. 2.3.1) and preserving neighbourhood relationships on the non-uniform grid (Sect. 2.3.2).

#### 2.3.1 Accommodating MRA process using Z-order curves

To efficiently perform MRA on the GPU, a data structure based on the Z-order curve is deployed, which maps the hierarchy of grids to a single 1D array. A Z-order curve can be created for a $2^l \times 2^l$ grid by following the so-called Morton codes of the grid elements, which are obtained by bit interleaving the x and y indices of each element. An example of the creation of a Z-order curve using the Morton codes for a $2^2 \times 2^2$ grid is shown in Figure 3. As seen in Figure 3a, X and Y indices are stored in binary form and bit interleaved, as indicated by the alternating red and black digits. After bit interleaving of the X and Y indices, the resulting Morton codes are converted to their equivalent decimal indices that are then joined in an ascending order to create the Z-order, as seen in Figure 3b.



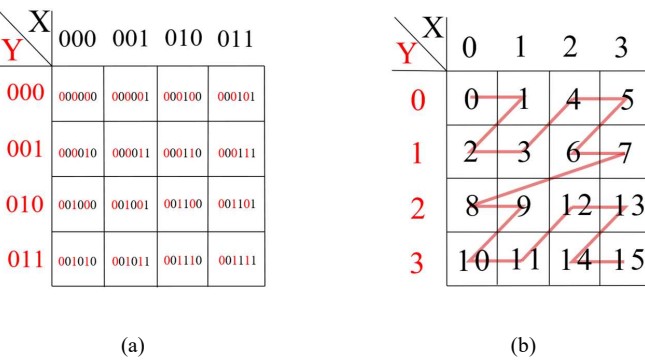

(a)                                          (b)

**Figure 3.** Indexing the elements of a grid using a Z-order curve: (a) bit interleaving the X and Y indices of each element, in binary form and (b) Morton codes for each element from interleaving, in decimal form.

Z-order curves are created for each grid in the hierarchy of grids to map the hierarchy into a single 1D array

residing in GPU memory. Figure 4a shows a hierarchy of grids with $L = 3$ that has been mapped to a 1D array using the Z-order curve. This indexing allows to efficiently perform the encoding process (e.g., the elements with indexes 9, 10, 11, and 12 that are required to get the details on element 2 all reside in adjacent locations in memory) and form the tree of significant details, as shown in Figure 4b.

After getting the tree of significant details, a number of GPU threads equal to the number of elements on the finest

grid, here $2^3 \times 2^3 = 64$ threads, are launched. All these threads start at the element with index 0 and climb the branches of the tree until they reach the leaf element and record it, as shown in Figure 4c. Depending on the resolution level of the leaf elements, some threads might record duplicate indices. These duplicate indices are removed to reduce the number of threads when assembling the non-uniform grid, as shown in Figure 4d.

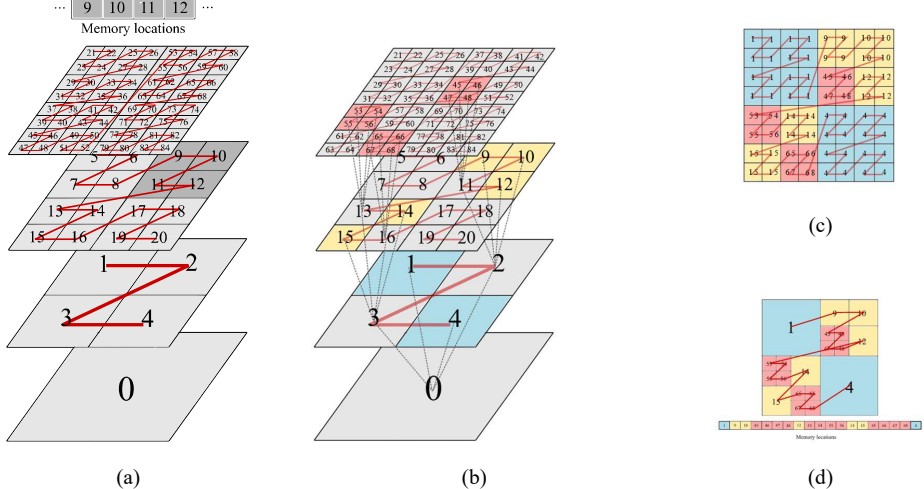

(a)                                    (b)                                    (d)

**Figure 4. (a) Indexing of each grid in the 3D hierarchy of grids using Z-order curves, (b) the tree of significant details indexed**
**using Z-order curves, (c) indices recorded by each GPU thread, including duplicate indices, (d) the Z-order curve of the final assembled non-uniform grid after removing duplicate indices.**

**2.3.2 Preserving neighbourhood relationships on the non-uniform grid**

The element-wise update using the non-uniform solver's scheme requires aggregating the discharges at each face it shares with the other neighbouring elements. Therefore, each element must find and store neighbour relationships about all of its





neighbours to be able to loop over the faces. These relationships can be stored by an equivalent representation of the non-uniform grid projected onto the finest grid, as shown in Figure 4c. In Figure 4c, element 1 is located at refinement level $l = 1$ and comprises 16 duplicate indices. It has 4 neighbours on the northern, eastern, southern and western sides. Considering the array of 16 duplicates, by knowing both current resolution level $l$ and the maximum refinement level $L$, the Morton codes of two points located at the southwest and northeast of the array can be found. Using these two Morton codes all the indices at

the four sides of element 1 can be identified while removing the duplicate indices to locate and save the actual neighbouring elements.

### 2.4 Updates to LISFLOOD-FP utilities for running the non-uniform

The non-uniform ACC solver is distributed as a new release for LISFLOOD-FP 8.1 and follows the same standard usage for the other solvers with a few updates to input/output components. In order to use the solver, first, the user needs to reshape

the DEM data as planar Galerkin projected data (recall Sect. 2.1). This is achieved by using a utility application called `generateDG2DEM`, which loads the raw DEM raster file and generates three new raster files containing the average (i.e., `.dem` format), $x$-slope and $y$-slope topography coefficients (i.e., `.dem1x` and `.dem1y` format). Configuring a specific simulation using the non-uniform ACC solver requires setting one additional parameter of the error threshold, $\varepsilon$, and extracting the maximum refinement level parameter, $L$, from the DEM (recall Sect. 2.1). To account for the rainfall, the

model follows the so-called rain-on-grid approach (Costabile et al., 2021), which simply imposes the prescribed rainfall data at grid elements. These prescribed rainfall data can be imported from a simple text file (i.e., `.rain` format) for spatially-uniform and temporally-varying rains, or from a NetCDF file (i.e., `.nc` format) for spatially- and temporally-varying rains. By default, the outputs of the new non-uniform ACC solver can be saved as multiresolution grids in VTK file format to enable the use of visualisation tools such as ParaView (https://www.paraview.org/) to plot the non-uniform grid and data.

Also, the outputs are further converted to raster files to allow benchmarking on uniform grids. Basic checkpoint/restart capabilities are available to resume computations from a previously saved snapshot. Starting with the 8.0 version, LISFLOOD-FP has transitioned to a new CMake-based build system, following the general trend in the open-source software community, which allows hassle-free building/compiling in a cross-platform manner. A step-by-step guide on how to download and install the code, along with instructions on simulating selected case studies using the new non-uniform

solver are given in a series of supplementary videos (Sharifian and Kesserwani, 2022), along with instructions on www.seamlesswave.com/LISFLOOD8.0.

### 3 Performance evaluation and benchmarking

The efficiency of the GPU versions of the uniform and non-uniform grid solvers are assessed by applying them to reproduce realistic fluvial and/or pluvial flooding scenarios. Five case studies are selected and investigated to demonstrate the potential

utility of the solvers for both catchment-scale and urban-scale flood simulations. The case studies are described in Table 1, which also includes information on the maximum refinement level, $L$, used to generate the grid with the non-uniform solver, and on the DEM resolution, $R$, at which the uniform solver was run. Non-uniform and uniform grid simulations were run on an Nvidia Quadro RTX 2070 SUPER GPU card. The uniform grid simulations were also run for the CPU version reported in Neal et al. (2018), on a 3.8 GHz Intel i7 8700 with 12 CPU threads, which led to identical predictions to those achieved on the GPU version. The closeness of the predictions from the non-uniform simulations to those of the uniform grid simulations

depends on the choice for the error threshold, $\varepsilon$. As this choice needs to be $\varepsilon \leq 10^{-3}$ to preserve an acceptable level of closeness (Kesserwani and Sharifian, 2020), the non-uniform solver simulations were run with $\varepsilon = 10^{-3}$ and $10^{-4}$. In this sense, the accuracy will hereafter refer to the level of closeness of the non-uniform solver simulations to those achieved by the uniform solver simulations on the finest resolution, $R$, that can be available on its grid.



**Table 1. Selected case studies used: to benchmark the accuracy of the non-uniform grid solver against the uniform grid solver, to analyse the speed-up gain from the uniform grid solver on the GPU architecture compared to the CPU architecture, and to also analyse the speed-up gain acquired by the non-uniform solver over the GPU version of the uniform grid solver.**

| Case studies | Source | Type | Elements* (in k) | $L$ | $R$ (m) | $T_s$ (hr) |
|---|---|---|---|---|---|---|
| Lower Triangle catchment | Özgen-Xian et al., 2020 | Pluvial | 149, 594, 3700 | 10, 11, 12 | 10, 5, 2 | 72 |
| Upper Lee catchment | Xia and Liang, 2018 | Pluvial | 2712 | 12 | 20 | 120 |
| Eden catchment | Xia et al., 2019 | Fluvial | 6276 | 13 | 20 | 132 |
| Glasgow urban area | Neelz and Pender, 2013 | Pluvial | 95 | 9 | 2 | 5 |
| Cockermouth urban area | Muthusamy et al., 2021 | Fluvial | 2160 | 11 | 1 | 144 |

\* The number of elements only involves the valid pixels of the DEM, excluding the ones without data.

The maximum flood extent predicted by the non-uniform grid solver is compared to that of the uniform solver using

the hit rate (H), false alarm (F) and critical success index (C) (Wing et al., 2017; Hoch and Trigg, 2019). The metric H measures how much the extent predicted by the uniform solver is covered by that predicted by the non-uniform solver (i.e., H = 1 indicates full coverage and 0 otherwise). The metric F measures how much of the extent the extent predicted by the non-uniform solver is outside that predicted by the uniform solver (i.e., F = 0 indicates that there are no predicted portions by the non-uniform solver that outside of the extent predicted by the uniform solver, and F = 1 indicates that extent predicted by

the former is entirely outside that predicted by the latter). The C metric combines H and F to weigh how much of the sum of both the extents predicted by the uniform and non-uniform solvers is covered by the extent predicted by the non-uniform solver (i.e., C = 1 indicates full coverage and 0 otherwise). Interpreting the C metric requires some judgement bespoke to each test case because large flooded areas with relatively short perimeters and fine grid resolutions can increase the score. Nevertheless, in many applications as C gets close to 0.8 the inundation extent prediction is often said to be sufficiently

accurate.

The speed-up gain associated with the use of the GPU version of the uniform grid solver, over the CPU version, will be quantified by looking at the ratio of their respective runtimes to complete a simulation. The enhancement in speed-up associated with the use of the non-uniform solver simulation over the uniform solver will be quantified similarly for the GPU versions. The grid predicted by the non-uniform solver entails a reduction in the number of elements with reference to the

grid of the uniform solver. Table 2 includes the simulation runtimes and the number of elements consumed by solvers for the selected case studies, from which the speed-up ratios and the aforementioned reduction will be quantified.

**Table 2. Simulation runtimes and number of elements on the grid for the uniform solvers (on CPU and GPU architectures) and for the non-uniform solver (GPU architecture) for all the selected case studies.**

| | Uniform solvers | | | Non-uniform solver at $\varepsilon = 10^{-3}$ | | Non-uniform solver at $\varepsilon = 10^{-4}$ | |
|---|---|---|---|---|---|---|---|
| | Number of elements (in k) | CPU time (in sec.) | GPU time (in sec.) | Number of elements (in k) | GPU time (in sec.) | Number of elements (in k) | GPU time (in sec.) |
| Lower Triangle catchment (Sect. 3.1) | | | | | | | |
| $R = 2$ m | 3700 | 35700 | 16640 | 60 | 764 | 272 | 2923 |
| $R = 5$ m | 594 | 2080 | 1257 | 24 | 165 | 109 | 410 |
| $R = 10$ m | 149 | 320 | 359 | 12 | 75 | 51 | 150 |
| | | | | | | | |
| Upper Lee catchment (Sect. 3.2) | | | | | | | |
| $R = 20$ m | 2712 | 55189 | 2557 | 726 | 820 | 2050 | 1960 |
| | | | | | | | |
| Eden catchment (Sect. 3.3) | | | | | | | |
| $R = 20$ m | 6276 | 202390 | 7333 | 984 | 620 | 4011 | 1944 |
| | | | | | | | |
| Glasgow urban area (Sect. 3.4) | | | | | | | |
| $R = 2$ m | 95 | 97 | 69 | 35 | 13 | 60 | 18 |
| | | | | | | | |
| Cockermouth urban area (Sect. 3.5) | | | | | | | |
| $R = 1$ m | 2160 | 71084 | 8019 | 871 | 7101 | 1362 | 9434 |



### 3.1 Lower Triangle catchment

The case study reproduces a real-world rainfall-runoff event over a mountainous catchment considering three DEM resolutions of 2, 5 and 10 m. It aims to assess the accuracy of the non-uniform grid solver with increasingly coarser DEM resolution. The catchment is a sub-catchment of the East River watershed in Colorado, USA, with a surface area of 14.84 km$^2$ featured by high-elevation topography, though with smoothly varying gradients (Figure 5). The variations are larger in the northwest and northeast corners and become smaller around the relatively flat main river stream located in the middle of

the catchment. The 72-hour flood event is induced by a storm, which occurred in August 2016, with rainfall time series given at 30-minute intervals (Figure 5). The friction parameter is uniformly set to $n_M = 0.035$ s/m$^{1/3}$ over the domain that was initially dry before a simulation starts. A simulation starts by loading the temporally-varying rainfall data, uniformly into all elements on the grid.

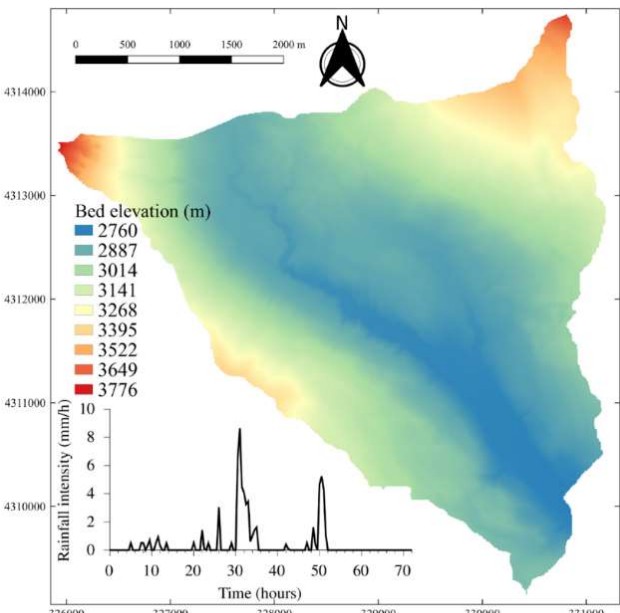

**Figure 5. Lower Triangle catchment. Topographic map of the domain with a subfigure showing the time series of the spatially-uniform rainfall over the catchment.**

The ranges of resolutions for the grids predicted by the non-uniform solver are compared in Figure 6, for the two $\varepsilon$ values and the three DEMs considered at 2, 5 and 10 m resolutions. With $\varepsilon = 10^{-4}$ and the 2 m resolution DEM, the grid predicted shows a dominance for the 8 m resolution as finer resolutions are barely reached. This leads to a 93 % reduction in

the number of elements and confirms that the DEM is featured by very smooth variations. With the 5 and 10 m resolution DEM, the reduction decreased to 82 % and 66 %, respectively, suggesting that the use of coarser DEM resolutions smoothens the representation of the topographic features. With $\varepsilon = 10^{-3}$, the grids predicted by the non-uniform solver show dominance of coarser resolutions ranging between 16 to 32 m, 20 to 40 m, and 40 to 80 m with the use of a 2, 5 and 10 m resolution DEM, respectively. In this case, there are higher than 92 % reductions in the number of elements with the 2, 5 and

10 m resolution DEMs, signalling that $\varepsilon = 10^{-3}$ is too big and the features of the topography are too smooth.






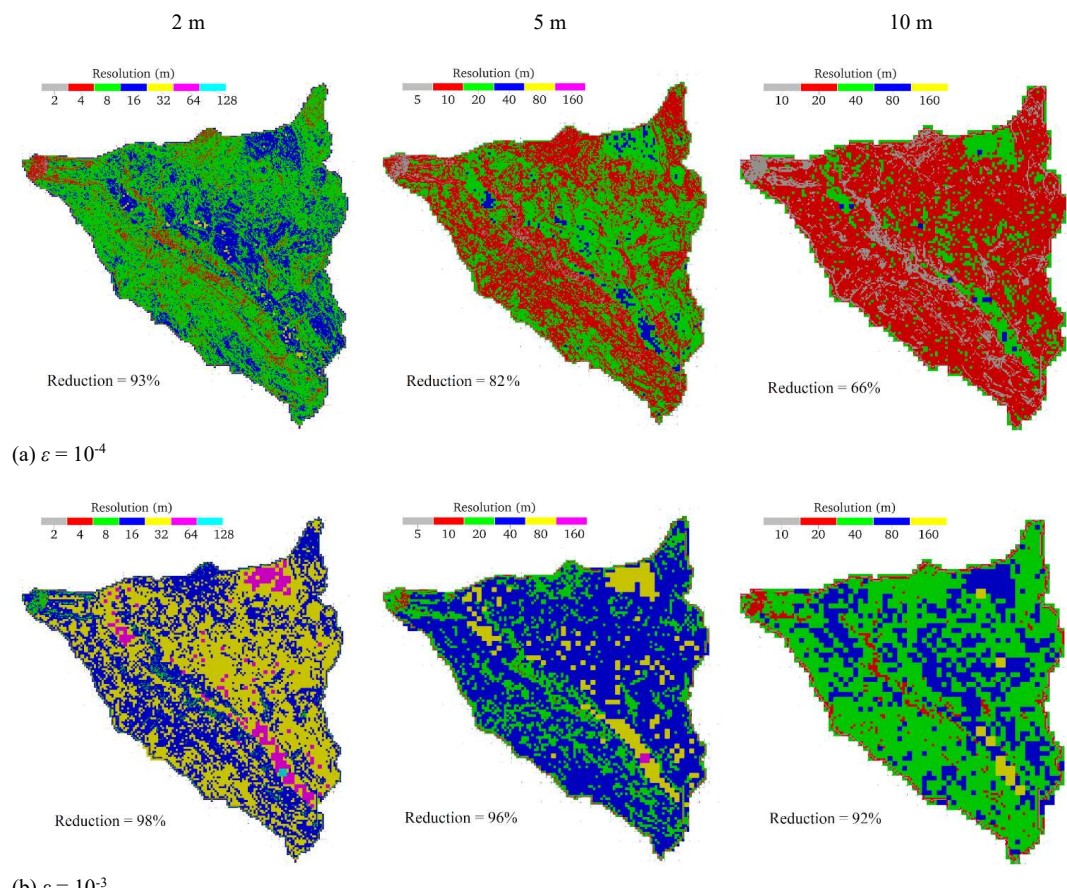

Figure 6. Lower Triangle catchment. The resolution maps for the grids generated by the non-uniform solver considering DEM at 2, 5 and 10 m resolutions, using: (a) $\varepsilon = 10^{-4}$ and (b) $\varepsilon = 10^{-3}$.

The accuracy of the non-uniform solver predictions of the flood extent is analysed in Figure 7, which shows the maps of the difference in maximum flood extents for the two $\varepsilon$ values. With $\varepsilon = 10^{-4}$, at 2 m DEM resolution, the predicted maximum flooded area shows the least agreement with the extent predicted by the uniform grid solver, with a C of 0.56, leading to pronounced underpredictions (red areas, H = 0.69) and overpredictions (blue areas, F = 0.24) over the tributary streams (Figure 7a). At the coarser 5 and 10 m DEM resolution, the overall agreement of the predictions by the non-uniform grid solver becomes close to the uniform grid counterparts, leading to higher C values of 0.61 and 0.67, respectively. The improvement in accuracy for the non-uniform solver predictions can be associated with the fact that there is less reduction in the number of elements on the non-uniform grid as the DEM resolution is coarsened (Figure 6a). The same tendency is observed for the larger $\varepsilon = 10^{-3}$, with a more pronounced loss of accuracy in maximum flood extent predictions due to the higher reductions in the number of elements on the non-uniform grid (Figure 6b), leading to C values of 0.43, 0.44 and 0.48 for 2, 5 and 10 m DEM resolutions, respectively (Figure 7b). However, the C values remain close to each other at $\varepsilon = 10^{-3}$, while remaining lower than 0.5, suggesting that this $\varepsilon$ value may be too large for simulations involving smooth DEMs at a resolution of 2 m or lower and should, therefore, be avoided (also shown next in Sects. 3.2 and 3.3).



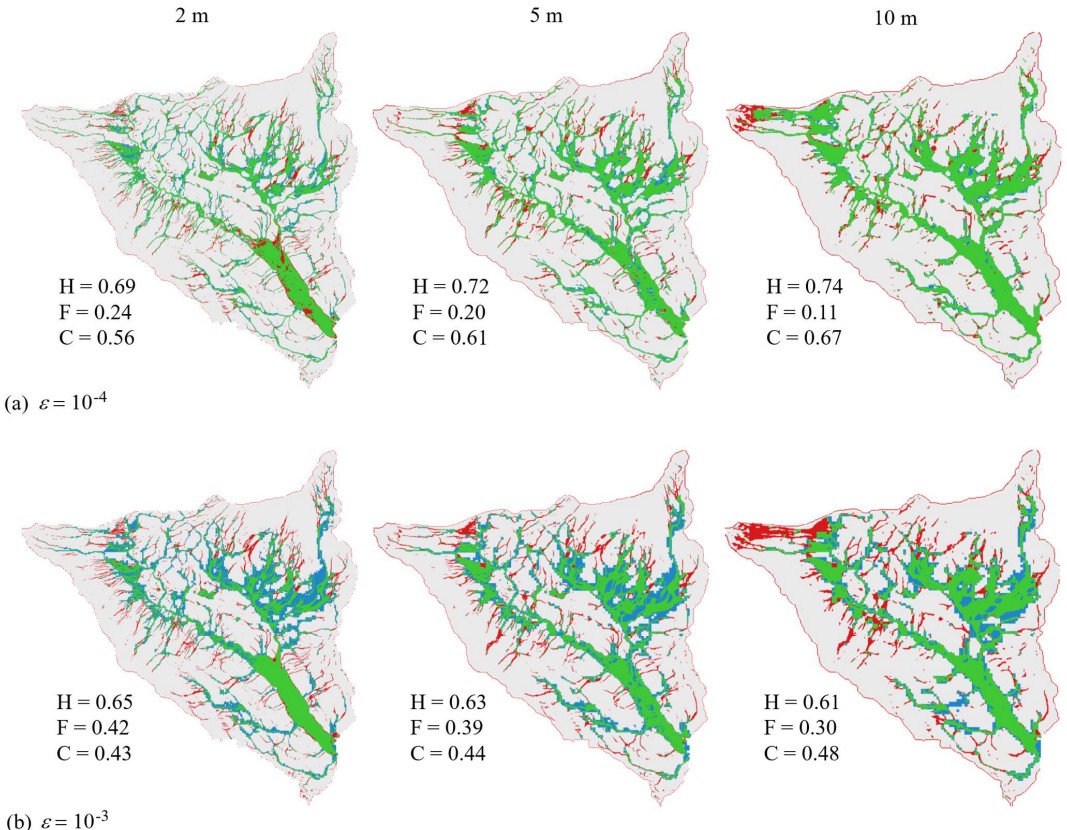

**Figure 7. Lower Triangle catchment. Difference in the maximum flood extent predicted by the non-uniform grid solver considering DEM resolutions of 2, 5 and 10 m (left to right), using: (a) $\varepsilon = 10^{-4}$; and (b) $\varepsilon = 10^{-3}$. Green parts flag flooded areas predicted by both the uniform and the non-uniform grid solvers, blue parts flag an overprediction (i.e., flooded areas only predicted by the non-uniform grid solver), red parts flag an underprediction (i.e., flooded areas only predicted by the uniform grid solver) and, white parts flag dry areas predicted by both the uniform and non-uniform solvers. The hit rate (H), false alarm (F) and critical success index (C) values are computed against the flood extents predicted by the uniform grid solver using respective DEMs.**

In terms of speed-up, the uniform solver on the GPU architecture is found to be 2.1, 1.65 and 0.9 times faster than the CPU architecture for the 2, 5 and 10 m DEM resolutions, respectively. The decreasing trend in speed-up ratios is expected as the number of elements decreases to less than 600 k and 200 k, for the 5 and 10 m DEM resolutions, respectively, for which the GPU parallelisation is not too effective in decreasing the runtimes (Shaw et al., 2021). With $\varepsilon = 10^{-4}$, the non-uniform solver leads to speed-up ratios of 5.7, 3.0 and 2.4 times over the uniform solver for the 2, 5 and 10 m DEM resolutions, respectively. Such an increasing trend in speed-up with increasingly finer resolutions is expected as there is a higher reduction in the number of elements (Figure 6a). With $\varepsilon = 10^{-3}$, the ratios become 21, 7.6 and 4.7 times, respectively. The significant speed-up achieved for the 2 m DEM resolution comes at the price of deteriorating accuracy (Figure 7). For the coarser 5 and 10 m DEM resolutions, the speed-ups are less than two and four times lower, respectively, but there is more deterioration in accuracy.

This case study indicates that the uniform grid solver on GPU architecture is more or less as fast as the CPU architecture when the number of elements is 500 k or smaller, but becomes increasingly faster when the simulation entails a





large number of elements. With $\varepsilon = 10^{-4}$, the non-uniform solver somewhat preserves a fair accuracy[1] and benefits from increasingly higher speed-ups as its grid entails increasingly higher reduction in the number of elements. With $\varepsilon = 10^{-3}$, the solver is unable to preserve an adequate level of accuracy when the resolution is coarse and the DEM is featured by smooth

topographic variations, suggesting to avoid the choice of $\varepsilon = 10^{-3}$ for such simulations.

### 3.2 Upper Lee catchment

This case study reproduces a rainfall-flood event in the Upper Lee catchment in north London, UK, which covers an area of 1180 km$^2$ shown in Figure 8a. The available DEM has a relatively coarse resolution, of 20 m, involving sharper topographic variations, compared to the previous case study (Sect. 3.1), with more defined tributaries and main river streams. This

catchment is also characterised by various classes of land use, shown in Figure 8b, which have been represented by spatially-varying Manning's friction parameter (Table 3) by following commonly recommended values (Chow, 1959). Heavy rainfall was recorded for 120 hours between the 5th and 11th Feb 2014, resulting in the flood event. Figure 8c shows the time series of the mean rainfall intensity that includes several rainfall peaks. The initial conditions for the water depth were pre-processed in a raster format to account for the fact that the rivers are initially wet before the simulation starts. The simulation

was then solely driven by the spatially- and temporarily-varying rainfall data. For this case study, an observed discharge hydrograph is available that was recorded at the gauge station located in the River Lee by the outlet of the catchment (Figure 8a). The accuracy of the non-uniform grid solver in the prediction of the flooding extent will be analysed for a smaller portion of the catchment (i.e., the framed portion in Figure 8a) that is located upstream of the gauge station from the north-eastern side. The impact on flooding extent predictions, over the whole catchment, will be analysed by comparing the solver

predictions to the peak flows with reference to the observed hydrograph.

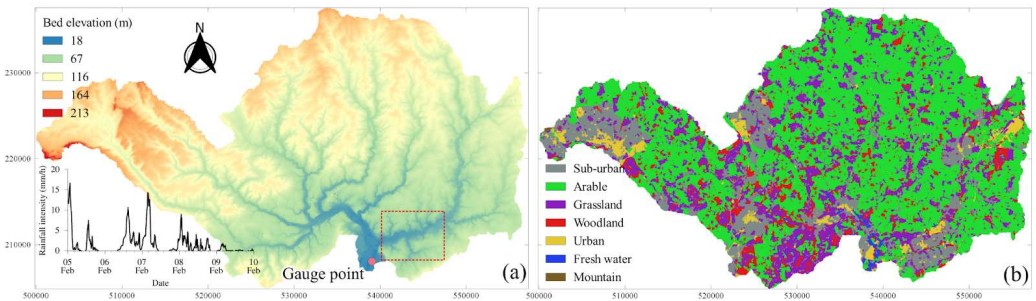

**Figure 8. Upper Lee catchment. (a) Topographic map of the domain including the position of the gauge point, a subfigure showing the time series of mean (spatially-varying) rainfall over the catchment, and a framed smaller portion of the domain in which the**
**maximum flood extent is analysed; (b) The land use map of the domain.**

**Table 3. Upper Lee catchment. Values of Manning's friction parameter for different classes of land use.**

| Land-use type | Sub-urban | Arable | Grassland | Woodland | Urban | Fresh water | Mountain |
|---|---|---|---|---|---|---|---|
| $n_M$ (s/m$^{1/3}$) | 0.13 | 0.125 | 0.075 | 0.16 | 0.03 | 0.015 | 0.15 |

Figure 9 compares the resolution maps generated from the grid predicted by the non-uniform solver for the two $\varepsilon$ values. With $\varepsilon = 10^{-4}$, the finest 20 m resolution is seen to cover about 70% of the catchment area to include the tributaries

converging into the main rivers, leading to an overall reduction of 25 % in the number of elements. With the larger $\varepsilon = 10^{-3}$, the grid is spanned by the 40 and 80 m resolutions that cover about 90 % of the catchment area. This leads to a higher reduction of 74 %, at the price of entailing overly coarser resolutions to represent the topographic connectivities.

---

[1] Exclusively in this case study, the C value does not approach 0.8. This may be expected due the shortcoming of the ACC solver's friction integration in the local areas of fast, thin water flows occurring over a very smooth and sloping terrain from a high intensity runoff (De Almeida et al 2012).



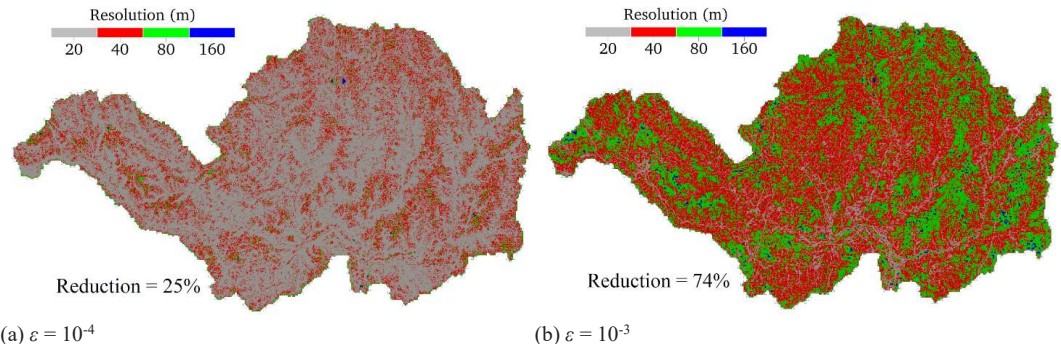

(a) $\varepsilon = 10^{-4}$        (b) $\varepsilon = 10^{-3}$

**Figure 9. Upper Lee catchment. The resolution maps for the grids generated by the non-uniform solver using: (a) $\varepsilon = 10^{-4}$ and (b) $\varepsilon = 10^{-3}$.**

320        Figure 10 compares the accuracy of the predictions made by the non-uniform grid solver for the two $\varepsilon$ values, in terms of maximum flood extent, for the selected portion of the catchment (Figure 8a). With $\varepsilon = 10^{-4}$ (Figure 10a), the solver accurately predicts the flood extent over the majority of the tributaries and main river streams, with rather high values for H and C metrics, of 0.86 and 0.80, respectively, and a low score for F of 0.07. Whereas, with $\varepsilon = 10^{-3}$, underpredictions can be detected mainly around the main river streams and their banks (red areas in Figure 10b, with lower H, of 0.80), given the

relatively coarser resolutions used to represent the topographic connectivity, leading to a reduction in overall accuracy with a lower C, of 0.71.

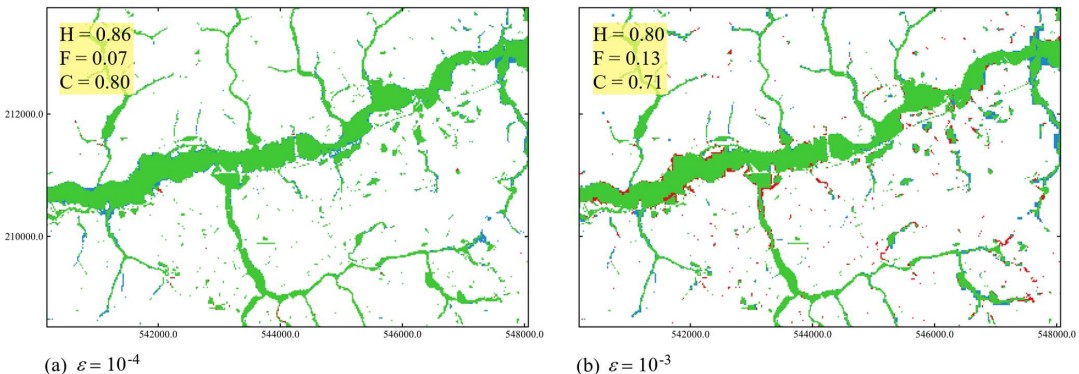

(a)  $\varepsilon = 10^{-4}$        (b)  $\varepsilon = 10^{-3}$

**Figure 10. Upper Lee catchment. Differences in the maximum flood extent predicted by the non-uniform grid solver using (a) $\varepsilon = 10^{-4}$ and (b) $\varepsilon = 10^{-3}$, over the smaller portion of the domain framed in Figure 8a. Green parts flag flooded areas predicted by both**

**the uniform and the non-uniform grid solvers, blue parts flag an overprediction (i.e., flooded areas only predicted by the non-uniform grid solver), red parts flag an underprediction (i.e., flooded areas only predicted by the uniform grid solver) and, white parts flag dry areas predicted by both the uniform and non-uniform solvers. The hit rate (H), false alarm (F) and critical success index (C) values are computed against the flood extent predicted by the uniform grid solver on 20 m resolution DEM over the whole domain.**

335        The predicted flow discharges at the outlet (post-processed hydrographs), are compared against the observed hydrograph in Figure 11. None of the simulated hydrographs closely trail the observed hydrograph. This deficiency can be attributed to uncertainties in the location of the in-situ measurements, in the aggregation of the simulated discharges at coarse resolutions, in the ability of Manning's friction law formula to model rain-driven overland flows at low Reynolds numbers (Kistetter et al., 2016; Taccone et al., 2020), or the fact that river channel bathymetry was excluded from the DEM.

Therefore, the predictability of the solvers to the flow discharge at the outlet is analysed by looking at the extent to which they can reproduce the primary and secondary peaks seen in the observed hydrograph. The uniform solver provides the closest trail to the primary peak, with a deviation of 9 %, and predicts the presence of the secondary peak at the falling limb (occurring between 80 and 90 hours). The non-uniform solver, with $\varepsilon = 10^{-4}$, although fairly predicts the rising limb,



underpredicts the primary peak with a larger deviation of 20 %, and overlooks the presence of the secondary peak. With $\varepsilon =$
$10^{-3}$, the solver leads to smeared predictions with a larger deviation of 38 %.

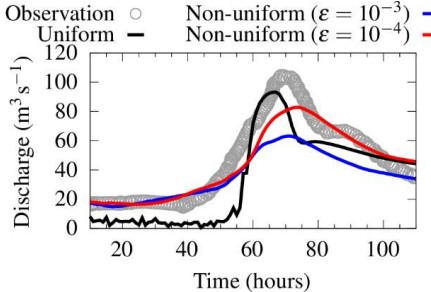

**Figure 11. Upper Lee catchment. Comparison of observed and simulated discharge hydrographs at the catchment outlet (marked in Figure 8a).**

In terms of speed-up, the uniform grid solver on GPU architecture is found to be 21 times faster to run than the
CPU architecture. This speed-up is significant, although the number of elements here is one million less than in the previous
case study for the 2 m resolution (Sect. 3.1). This may be attributed to the higher intensity and longer duration of the rainfall
event featuring this case study, leading to a higher number of elements that actually get inundated during the simulation.
With $\varepsilon = 10^{-4}$, the non-uniform grid solver offers a marginal speed-up of 1.3 times, which is increased to 3.1 times with the
larger $\varepsilon = 10^{-3}$; this is expected as the reduction in the number of elements increases from 25 % to 74 %, respectively (Figure
9). This retrieves the previous observation (Sect. 3.1), suggesting that the speed-up afforded by the non-uniform grid solver
is proportional to the reduction in the number of elements. Though the reduction is much higher with $\varepsilon = 10^{-3}$, this value is
again deemed impractical for such a coarse resolution catchment-scale simulation; it would in fact allow for more coarsening
to a DEM resolution that is already as coarse as 20 m, which impacts accuracy. For such a simulation with the non-uniform
solver, choosing $\varepsilon = 10^{-4}$ is necessary to keep a moderate level of coarsening in favour of accuracy, while expecting
considerable speed-ups over the uniform solver when the reduction in the number of elements exceeds 30 % (shown next in
Sect. 3.3).

### 3.3 Eden catchment

This case study replicates the rainfall-driven fluvial flooding over the Eden catchment in northwest England, caused by
Storm Desmond in December 2015. Figure 12a shows the 2500 km$^2$ catchment, including rivers Irthing, Petteril Caldew and
Eden, which form a confluence in the city of Carlisle (Figure 12b). The storm event was previously stimulated for DEM
resolutions ranging from 40 m to 5 m, with a variety of uniform grid solvers formulated upon different levels of
mathematical and numerical complexity (Xia et al., 2019; Ming et al., 2020; Shaw et al., 2021). In favour of computational
efficiency, these studies recommend using the 10 m DEM resolution as it is sufficient to replicate the flood flow dynamics
around the urban areas with enough details. Moreover, the uniform ACC solver could capture the required flow features at
the coarser 20 m DEM resolution with minor differences (see Figure 15 in Shaw et al., 2021). For this case study, the 20 m
DEM resolution was used to demonstrate the performance of the non-uniform grid solver. This is because the latter could not
be run for the 10 m DEM resolution, hampered by the memory costs for the non-uniform grid generation for this large
catchment, or in other words by the fact that the number of elements cumulated on the grid hierarchy exceeded the capacity
of the available GPU cards.


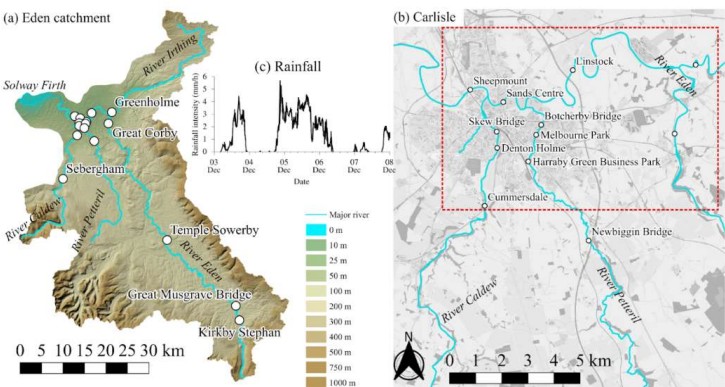

**Figure 12. Eden catchment. Topographic map of: (a) the catchment area covering 2500 km²; (b) the area over the city of Carlisle at the confluence of rivers Irthing, Petteril, Caldew and Eden. Names and locations of the 16 sampling points are marked, along with the dashed red line framing a smaller portion of the domain where the maximum flood extent was surveyed. Panel (c) shows the time series of mean (spatially-varying) rainfall over the catchment. © Google Maps 2021.**

A simulation starts at 00:00, 3 December 2015, with a 36-hour spin-up from an initially dry domain, and ends at 12:00, 8 December 2015. The spatially- and temporally-varying rainfall is shown in Figure 12c, which is provided from radar data at a 1 km spatial resolution and 5 min temporal resolution (Met Office, 2013). The rainfall file, in NetCDF file format, is loaded to start the simulation. The friction parameter is set to $n_M = 0.035$ s/m$^{1/3}$ for river channels and $n_M = 0.075$ s/m$^{1/3}$ for floodplains. The water levels simulated by the uniform and non-uniform grid solvers were recorded in 16 gauge points located in river channels (Figures 12a and 12b) to enable comparison with in-situ measurements (Environment Agency, 2020). The maximum predicted flood extents were also recorded to enable comparison with a surveyed post-event flood extent in the portion inside the Carlisle area framed in Figure 12b (McCall, 2016).

Figure 13 shows the range of resolutions predicted on the non-uniform grids for the two $\varepsilon$ values. With $\varepsilon = 10^{-4}$, it can be seen that the finest 20 m resolution covers more than 50 % of the catchment area, or otherwise the 40 m resolution dominates. The larger $\varepsilon = 10^{-3}$ leads to 90 % dominance of the 40 and 80 m resolutions over the catchment area, with almost an equal distribution for the 80 m resolution. This reinforces that $\varepsilon = 10^{-3}$ allows more coarsening than necessary for a DEM resolution as coarse as 2 m (recall Sect. 3.2). The reduction in the number of elements is found to be significantly larger with $\varepsilon = 10^{-3}$, to be around 85 %, compared to the 36 % reduction obtained with $\varepsilon = 10^{-4}$ for which the topographic connectivities in the catchment were represented at the recommended resolutions (Xia et al., 2019; Shaw et al., 2021; Ming et al., 2020).

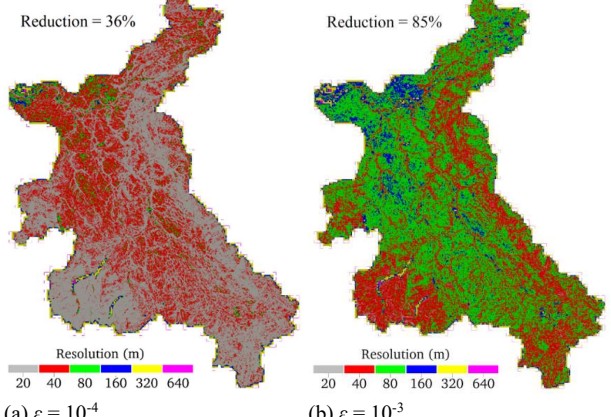

(a) $\varepsilon = 10^{-4}$      (b) $\varepsilon = 10^{-3}$

**Figure. 13. Eden catchment. The resolution maps for the grids generated by the non-uniform solver using: (a) $\varepsilon = 10^{-4}$ and (b) $\varepsilon = 10^{-3}$.**





In Figure 14, the simulated water level hydrographs are compared to the observations at the 16 gauge points. Expectedly, the hydrographs extracted from the uniform grid simulation show the closest agreement with the observed hydrographs, correctly capturing the rising and falling limbs and the peak water levels, in particular at the gauge points where the 20 m resolution is sufficient (Shaw et al., 2021). These points include Sheepmount, Sands Centre, Linstock and Great Corby, located in river channels whose widths are 2 to 3 times larger than 20 m (i.e., ranging between 54 and 71 m). The other points are located in river channels whose widths are close to, or less, than 20 m (i.e., ranging between 8 and 33 m), showing relatively a poorer prediction of the rising and falling limbs. The underprediction observed at Great Musgrave occurred because of an anomaly from the localised terrain elevation differences between the finest-resolution DEM and riverbed elevation measurements (Xia et al., 2019).

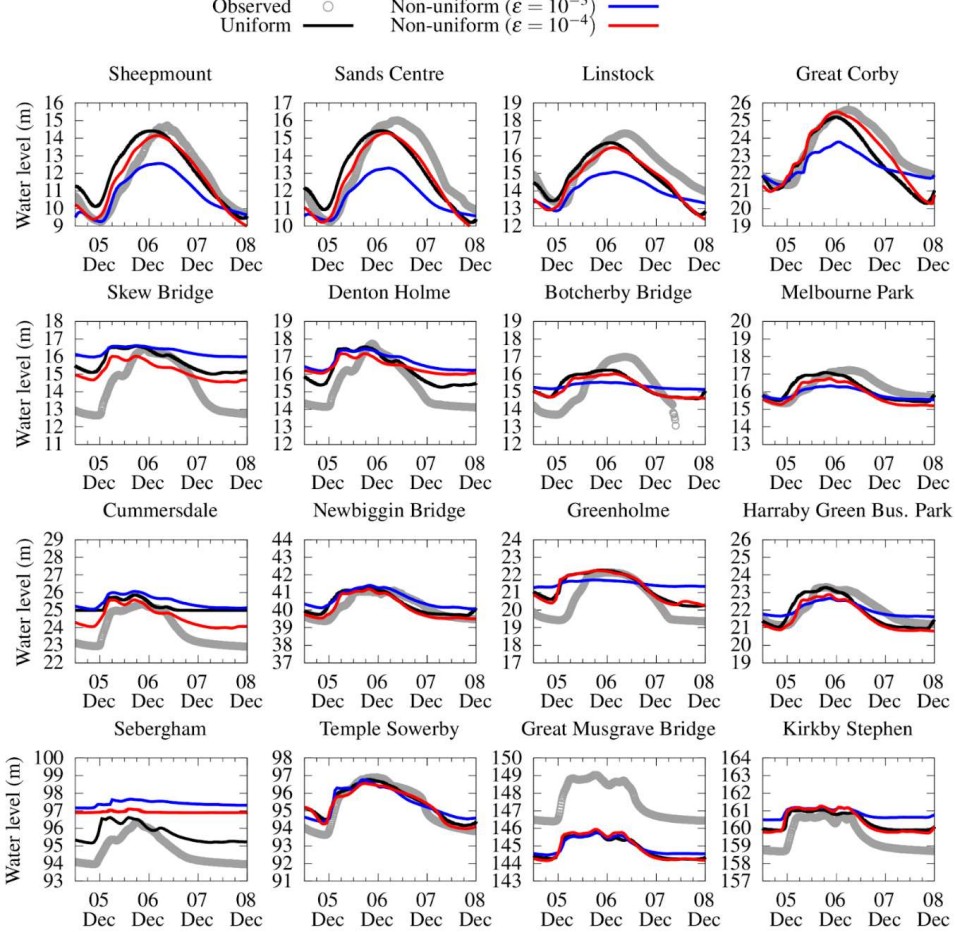

**Figure 14. Eden catchment. Water level time series predicted by uniform and non-uniform solvers and the in-situ measurements at 16 gauge points (marked in Figure 12).**

In terms of accuracy, the non-uniform solver with $\varepsilon = 10^{-4}$ predicts close water levels to the reference predictions by the uniform solver at almost all the gauge points. With the large $\varepsilon = 10^{-3}$, the non-uniform solver leads to a less accurate trail of the reference water levels, such as underpredictions to the peak levels at Sheepmount, Sands Centre, Linstock and Great Corby alongside smearing of the hydrograph profiles at Sebergraham, Greenholm and Botchery Bridge. This loss in accuracy is caused by the poorer representation of the topographic features excluding river streams on a grid dominated by the 80 m resolution (Figure 13), which again reinforces the impracticability of using $\varepsilon = 10^{-3}$ with the non-uniform grid solver for a catchment-scale simulation at resolutions as coarse as 20 m (recall Sect. 3.2).



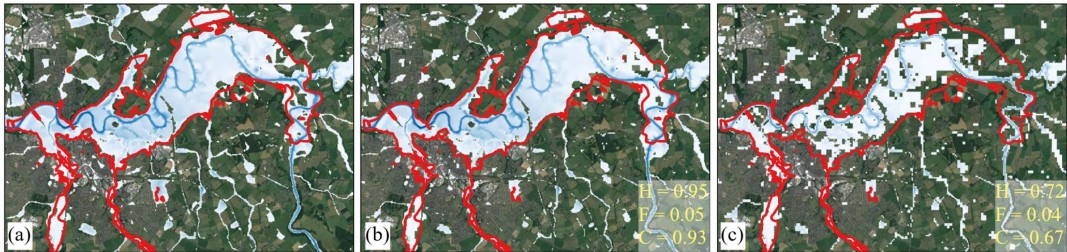

**Figure 15. Eden catchment. Maximum flood extents predicted by (a) uniform grid solver, (b) non-uniform grid solver using $\varepsilon = 10^{-4}$, and (c) non-uniform grid solver using $\varepsilon = 10^{-3}$, along with the measured flood extent from the Environment Agency bordered by the red line. The hit rate (H), false alarm (F) and critical success index (C) values are computed against the flood extent from the uniform grid solver. © Google Earth 2021.**

The impact on accuracy associated with this choice for $\varepsilon$ can also be clearly noted in the comparison of the maximum predicted flood extents to the surveyed extent (Figure 15). As can be seen in Figure 15a, the uniform grid solver predicts extents that compare reasonably well with the surveyed extent. With $\varepsilon = 10^{-4}$, The non-uniform grid solver delivers a competitive prediction to flood extent compared to the uniform grid solver as it achieved very close scores for H, C, and F, of 0.95, 0.93 and 0.05, respectively. These scores became significantly lower with $\varepsilon = 10^{-3}$, in particular for H and C, of 0.72 and 0.67, indicating a 25 % loss in accuracy of the flood extent, most visible in the underprediction of the easternmost side of the city area (Figure 15c).

In terms of speed-up, the uniform grid solver on GPU architecture is found to be 28 times faster to run than the CPU architecture. This speed-up is somewhat one order-of-magnitude higher than for the previous case study (Sect. 3.2), which is expected as the number of elements on the uniform grid here is more than double (i.e., 6.2 million versus 2.7 million for Upper Lee catchment). With $\varepsilon = 10^{-4}$, the non-uniform grid solver adds on a speed-up of around 4 times, associated with a 36 % reduction in the number of elements; whereas, with $\varepsilon = 10^{-3}$ the reduction is 85 % and this further boosts the solver's speed-up to around 12 times. Nonetheless, choosing $\varepsilon = 10^{-4}$ is necessary with the non-uniform grid solver for such a large-scale simulation at a 20 m DEM resolution to keep acceptable accuracy while benefiting from a considerable speed-up.

Overall, the uniform grid solver on GPU architecture is a promising alternative to boost runtime efficiency for catchment-scale flood simulations that entail large numbers of elements. This efficiency gain can be further enhanced by the use of the non-uniform grid solver, albeit with $\varepsilon = 10^{-4}$ for the range of DEM resolution used for such simulations to the resolution coarsening moderate in favour of accuracy. The amount of enhancement in efficiency depends on the reduction in the number of elements, compared to the uniform grid, becoming worth consideration when this reduction is 30 % or higher.

### 3.4 Glasgow urban area

This case study mimics an urban flood event in a 400 m × 960 m area in Glasgow, UK, shown in Figure 16. It is a hypothetical scenario with a DEM at 0.5 m resolution that excludes the fine-scale urban features. Therefore, a 2 m resolution DEM is often used to benchmark flood modelling softwares (Neelz and Pender, 2013). The flood is driven by rainfall that overwhelms the drainage system. This causes two sources of flooding, including from spatially-uniform rainfall occurring between 1 and 4 minutes, with an intensity of 400 mm/h. The other source is at the location denoted by "S" in Figure 16, where flow surcharges from the drainage system, during 23 and 55 minutes, reaching a peak of 5 m³/s, at t = 37 minutes. The area has two different classes of land use: roads and pavements and other land characteristics represented by $n_M = 0.02$ s/m$^{1/3}$ and $n_M == 0.05$ s/m$^{1/3}$, respectively. The 5-hour simulation starts over an initially dry area, considering closed boundary conditions. The time series of predicted water levels were recorded at 9 gauge points which are shown in Figure 16. As no



observation data are available, the predictions from an alternative uniform grid simulation at the 0.5 m resolution were used as reference.

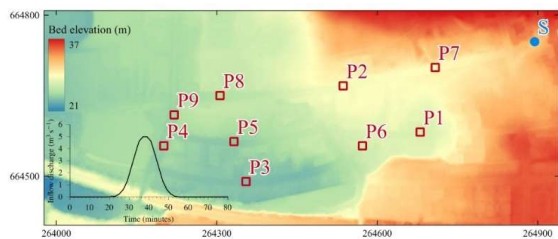

**Figure 16. Glasgow urban area. Topographic map of the domain, including the positions of the gauge points (P1-P9) and the source inflow (S), with a subfigure showing the inflow hydrograph imposed at point S.**

The grids predicted by the non-uniform solver for the two $\varepsilon$ values are illustrated in Figure 17 (upper panels), together with subfigures showing the percentage of coverage for each of the resolutions selected over the area. The finest 2 m resolution is selected in the vicinity of the southern boundary for the two $\varepsilon$ values, covering about 20 % and 50 % of the

area with $\varepsilon = 10^{-3}$ and $10^{-4}$, respectively. As the DEM lacks fine-scale urban features, the 4 m resolution is selected to cover the majority of the area almost equally with $\varepsilon = 10^{-3}$ and $10^{-4}$, at 60 % and 55 %, respectively. Notably, with $\varepsilon = 10^{-3}$, there is more presence of coarser than 4 m resolutions leading to an overall reduction of 64 % in the number of elements, compared to with $\varepsilon = 10^{-4}$ that barely uses coarser than 4 m resolution leading to a less reduction of 36 %. This suggests that $\varepsilon = 10^{-3}$ may be suitable for the non-uniform solver for the 2 m resolution DEM to moderately coarsen its grid while benefiting from

a higher reduction for efficiency.

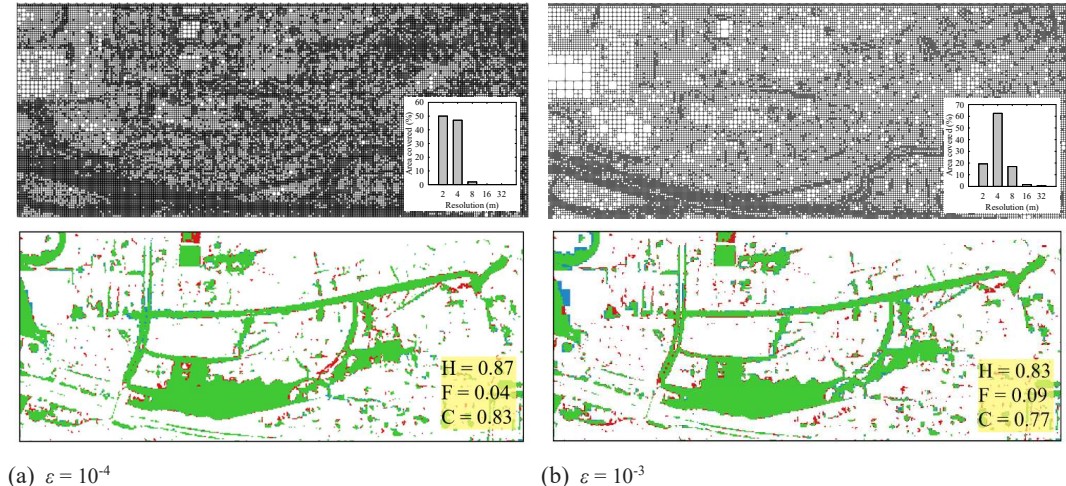

(a) $\varepsilon = 10^{-4}$   (b) $\varepsilon = 10^{-3}$

**Figure 17. Glasgow urban area. Grids generated by the non-uniform solver (upper panels), along with the percentage of the selected resolutions over the urban area; and the differences in the maximum flood extent predicted by the non-uniform grid solver (lower panels), using (a) $\varepsilon = 10^{-4}$ and (b) $\varepsilon = 10^{-3}$. In the lower panels, green parts flag flooded areas predicted by both the uniform and the non-uniform grid solvers, blue parts flag an overprediction (i.e., flooded areas only predicted by the non-uniform**
**grid solver), red parts flag an underprediction (i.e., flooded areas only predicted by the uniform grid solver) and, white parts flag dry areas predicted by both the uniform and non-uniform solvers. The hit rate (H), false alarm (F) and critical success index (C) values are computed against the flood extent predicted by the uniform grid solver on 2 m resolution DEM.**

Figure 17 (lower panels) also compares the accuracy of the non-uniform grid solver with both $\varepsilon$ values for the

maximum flood extent predictions. The solver with $\varepsilon = 10^{-3}$ and $10^{-4}$ leads to close predictions with only a noticeable difference in the vicinity of the western boundary where it overpredicts the flood extent with $\varepsilon = 10^{-3}$ (i.e., blue areas, with slightly higher F of 0.09), as compared to with $\varepsilon = 10^{-4}$, which is expected due to relatively coarser resolution therein. The remaining flooded areas are closely predicted with the two $\varepsilon$ values (i.e., large green areas), with rather high C values of 0.87



and 0.83, for $\varepsilon = 10^{-4}$ and $\varepsilon = 10^{-3}$, respectively. This confirms that the additional coarsening achieved by the non-uniform
grid solver with $\varepsilon = 10^{-3}$ only introduces an insignificant loss of accuracy.

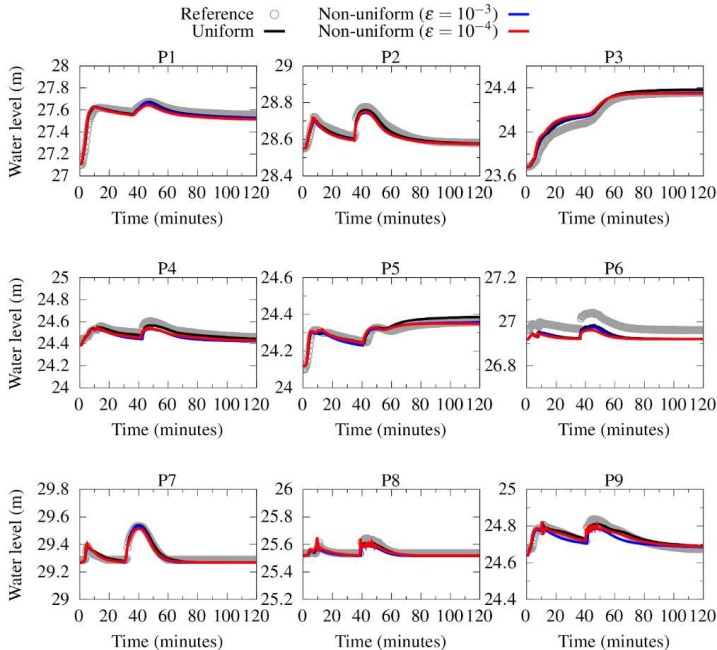

**Figure 18. Glasgow urban area. Water level time series predicted by uniform and non-uniform solvers and reference water levels predicted by the uniform solver on a 0.5 m × 0.5 m grid, at gauge points 1 to 9 (marked in Figure 16).**

This claim can be confirmed by comparing the results in Figure 18, which shows the time series of the water levels
predicted by the uniform and non-uniform grid solvers against the reference predictions. At all the gauge points, the non-
uniform solver, with both $\varepsilon = 10^{-3}$ and $10^{-4}$, leads to almost similar accuracy while being able to reproduce the double peak
observed for the reference predictions at 0.5 m resolution, in line with alternative predictions made by a variety of industry-
standard flood modelling software, including MIKE FLOOD, TUFLOW and ISIS 2D (Neelz and Pender, 2013).

In terms of speed-up, the uniform grid solver on GPU architecture is only 1.4 times faster than the CPU version,
which is expected for a number of elements as low as 95 k as discussed previously (Sect. 3.1). With $\varepsilon = 10^{-4}$, the non-
uniform grid solver boosts the speed-up to 3.8 times, which in turn increases to 5.3 times with $\varepsilon = 10^{-3}$ which entails a higher
reduction in the number of elements. All considered, $\varepsilon = 10^{-3}$ is found to be a valid choice with the non-uniform grid solver
for urban resolution flood simulations to both preserve acceptable accuracy and boost efficiency. The relevance of this
choice is further investigated next for a realistic case study with an urban resolution DEM that actually includes the fine-
scale features.

### 3.5 Cockermouth urban area

The case study involves a 144-hour fluvial flooding scenario over a 1.42 km$^2$ urban area in Cockermouth town, Cumbria,
UK. The town is located at the confluence of river Derwent and its tributary, River Cocker (Figure 19). Like the Eden
catchment (Sect. 3.3), this area was affected by the record-breaking rainfall from Storm Desmond in 2015 that manifested in
a water level increase of the River Derwent that caused river waters to inundate the town (Met Office, 2018). Measured time
series of the volumetric flow at 15 min intervals (from 03/12/2015 to 09/12/2015) were available at Southwaite Bridge and
Ouse Bridge for the River Cocker and River Derwent, respectively (Environment Agency, 2018). As these two locations are
considerably farther from the boundaries, the inflow hydrographs at Rivers Cocker and Derwent were calibrated




(Muthusamy et al., 2021) and imposed (at the two associated points shown in Figure 19). Here, the 1 m resolution DEM is

inclusive of the fine-scale urban features. Two different values were used for the friction parameter, $n_M$ = 0.035 and 0.05 s/m$^{1/3}$ for river channels and floodplains, respectively. The northern, southern and eastern boundaries are set as closed, while a free outflow boundary condition is allowed at the downstream of river Derwent. There are two gauge points on the rivers, located at P1 (River Cocker) and P2 (River Derwent), marked in Figure 19, where water depths predicted by the solvers were recorded to compare with existing in-situ measurements.

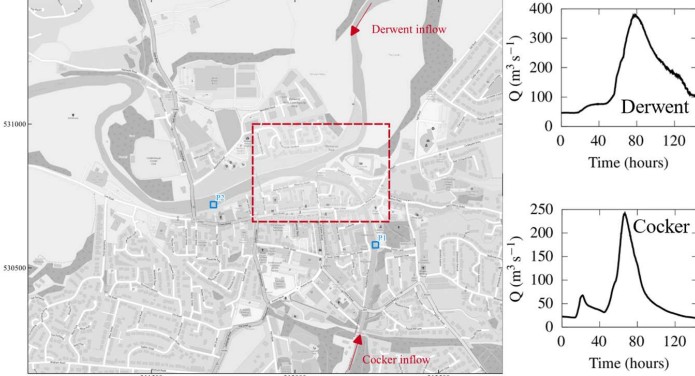


**Figure 19. Cockermouth urban area. The study area and the positions of the gauge points P1 and P2. The arrows show the positions where the inflow hydrographs are imposed at the upstreams of Rivers Derwent and Cocker. The dashed line frames a smaller portion of the domain where the generated grids are compared. © OpenStreetMap contributors 2021. Distributed under the Open Data Commons Open Database License (ODbL) v1.0.**

Figure 20 compares the generated non-uniform grids using two $\varepsilon$ values over the smaller portion of the domain framed in Figure 19, along with subfigures showing the percentage of the resolutions covering the whole study area. As shown in Figure 20a, the choice of $\varepsilon = 10^{-4}$ leads to an overly refined grid, where over 50 % of the area is covered with the finest 1 m resolution and entailing a 37 % reduction in the number of elements. Comparatively, as shown in Figure 20b, $\varepsilon = 10^{-3}$ leads to a more sensible resolution selection as the finest 1 m resolution is only present around the urban features (e.g.,

the buildings), while up to three-level coarser resolutions, i.e., 8 m, are selected over the pathways, and even coarser resolutions, between 16 and 32 m, are selected over the river channels. This leads to a reduction of 60 % associated with a wider range and more evenly distributed resolution selection compared to the smaller choice of $\varepsilon = 10^{-4}$ (see the subfigures in Figure 20).

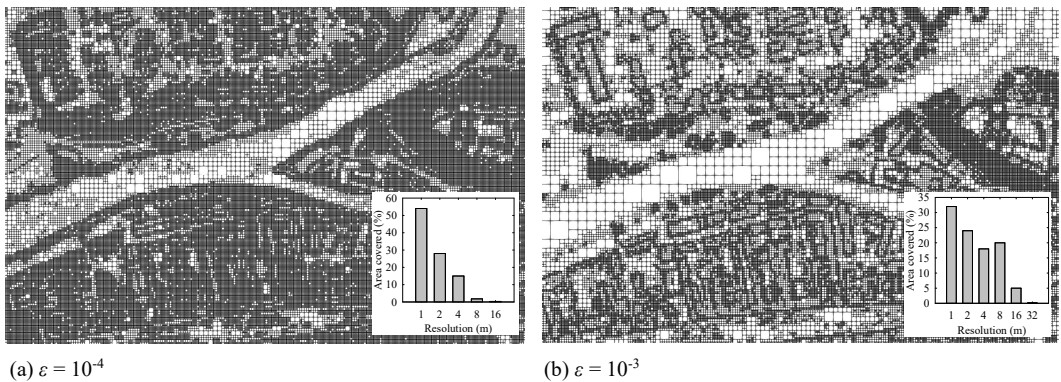

(a) $\varepsilon = 10^{-4}$                                                                (b) $\varepsilon = 10^{-3}$

**Figure. 20. Cockermouth urban area. Grids generated by the non-uniform solver using (a) $\varepsilon = 10^{-4}$ and (b) $\varepsilon = 10^{-3}$, over the**
**smaller portion of the domain framed in Figure 19, along with the percentage of the selected resolutions over the whole urban area.**



Figure 21 assesses the accuracy of the non-uniform grid solver with both $\varepsilon$ values in terms of water depth hydrograph predictions at gauge points P1 and P2 (Figure 21a-b) and maximum flood extent (Figure 21c-e) predictions, compared to the measured hydrographs and the surveyed flood extent, respectively. As can be seen in Figures 21a and 21b, the hydrograph predictions by the uniform grid solver follow the rising limb of the measured hydrograph up to the peak of the hydrograph at 70 hours, where it only shows a 20 cm underprediction. Also, the uniform grid solver could correctly capture the falling limb, with a slightly more gradual decrease from the measured hydrographs. The hydrographs predicted by the non-uniform grid solver with the two $\varepsilon$ values are very close to those predicted by the uniform solver with a negligible difference.

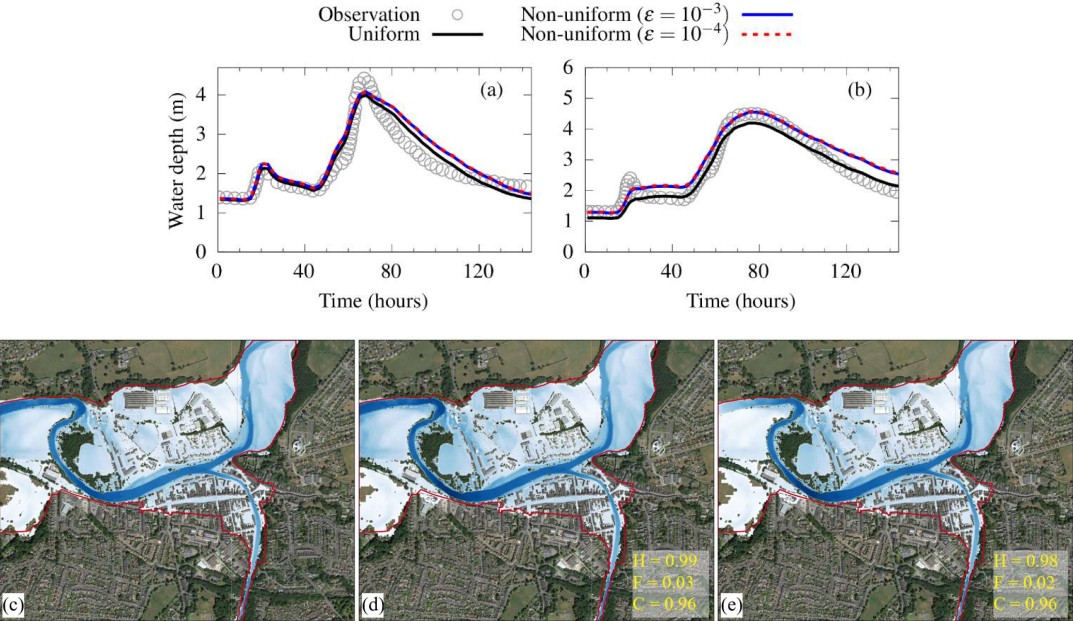

**Figure 21. Cockermouth urban area. Water level time series predicted by uniform and non-uniform solvers compared to the in-situ measured water levels at locations (a) P1 (River Cocker) and (b) P2 (River Derwent); along with maximum flood extents predicted by (c) uniform grid solver, (d) non-uniform grid solver using $\varepsilon = 10^{-4}$, and (e) non-uniform grid solver using $\varepsilon = 10^{-3}$. The measured flood extent from the Environment Agency is denoted by the red line. The hit rate (H), false alarm (F) and critical success index (C) values are computed against the flood extent from the uniform grid solver. © Google Earth 2021.**

The equally good accuracy for the non-uniform solver predictions with the two $\varepsilon$ values can also be confirmed by their maximum flood extents (Figures 21d and 21e) which are visually comparable to the one predicted by the uniform grid solver (Figure 21c), all agreeing well with the surveyed extent (Environment Agency, 2018). The accuracy of the non-uniform solver prediction can be quantitatively confirmed by computing the H, F and C metrics (defined in Sect. 3.3) with reference to the predicted maximum flood extent by the uniform solver. The scores for H, F and C achieved by the non-uniform grid solver with $\varepsilon = 10^{-4}$ and $\varepsilon = 10^{-3}$ are very similar, with a C value exceeding 0.9, suggesting close accuracy in the flood extent prediction.

In terms of speed-up, the uniform grid solver on GPU architecture ran 8.9 times faster than the CPU architecture. This is in line with the previous findings (Sects. 3.1 to 3.3), as more than one million elements were involved in the grid for this case study, confirming the utility of the GPU architecture to run faster simulations with a large number of elements. However, this gain in speed-up can get compromised when using the non-uniform grid solver. With $\varepsilon = 10^{-4}$, the non-uniform grid solver is in fact a bit slower (1.1 times) than the uniform grid solver, which may be expected as the grid is dominated by the finest resolution (Figure 20a) with 37 % reduction. Remarkably, with $\varepsilon = 10^{-3}$, which led to a much coarser grid with a 60 % reduction, the speed-up is insignificant, only 1.2 times faster than the uniform grid solver. This little gain in



speed-up could be explained by contrasting the depth, variability, and distribution of the resolutions selected on the non-
uniform grids to those in the other case studies (Sects. 3.1 to 3.4). It can be noted that the non-uniform grids in the other case
studies are featured by the dominance of one or at most two resolutions, depending on the choice for $\varepsilon$. This means that the
non-uniform grid solver has more access to regularly sized neighbours, which helps maximise its efficiency. In contrast, the
grids here are featured by more depth and variability in resolution, selected with a quite even distribution, in particular with $\varepsilon$
$= 10^{-3}$ (Figure 20b). This culminates in a high number of differently-sized elements, causing an overhead in the solver
calculation at those elements surrounded by too many neighbouring elements with different sizes. Such a knock-on effect on
the efficiency of the non-uniform solver would be expected to increasingly reduce as the modelled area of interest is larger
and more inclusive of rural surroundings at coarser resolutions. Still, the speed-up analysis confirms that the non-uniform
solver with $\varepsilon = 10^{-3}$ is a good choice for urban-scale simulations, to keep close accuracy to the uniform solver on GPU
architecture and remain more efficient. As the modelled area of interest becomes smaller and the terrain features are well-
resolved in the DEM, the efficiency gain over the uniform solver may not be substantial and this makes it a competitive first
choice for such a type of urban-scale simulations.

## 4 Conclusions and recommendations

The two-dimensional ACC uniform grid solver has been widely used to run efficient simulations for a wide range of flood
modelling applications. The efficiency of the ACC solver is owed to a reduced form of the shallow water equations solved
by a simple numerical scheme, and optimisation measures for the Central Processing Unit (CPU) architecture with effective
handling of wet-dry zones. This paper presented two potentially faster versions that run on the Graphical Processing Unit
(GPU): a uniform grid version built on the existing codebase (Shaw et al. 2021), and a non-uniform grid version within an
ad-hoc codebase with guidance on how to set-up and run these new versions on www.seamlesswave.com/LISFLOOD8.0.

       The non-uniform solver has a grid generator that uses the multiresolution analysis (MRA) of the multiwavelets to a
planar Galerkin projection of the digital elevation model (DEM), while only requiring one user-specified parameter as error
threshold, $\varepsilon$. This allowed to sensibly coarsen resolutions and to properly distribute different classes of Manning's
parameters at the coarsened elements. The ACC solver's scheme is adapted to aggregate the discharges across non-uniformly
sized grid elements. Both the grid generator and the adapted scheme are implemented in a new codebase to efficiently run on
the GPU architecture.

580        The performance of the GPU versions of the uniform and non-uniform grid solvers is assessed for five flooding
case studies at the catchment- and urban-scale. The non-uniform solver simulations were run for two $\varepsilon$ values, $10^{-3}$ and $10^{-4}$,
according to which accuracy was measured in terms of prediction closeness to those of the uniform solver run at DEM
resolution. The speed-up gained by using the GPU version of the uniform solver, over the CPU version, was quantified using
the ratio of runtimes with reference to the number of elements on the grid. Similarly, the enhancement in speed-up from
using the non-uniform solver over the uniform solver on the GPU was quantified with reference to the relative reduction in
the number of elements on the non-uniform grid.

       For the catchment-scale simulations at DEM resolution as coarse as 10 m or higher, the non-uniform solver with $\varepsilon$
$=10^{-4}$ generated grids without excessive resolution coarsening leading to acceptable accuracy. For such simulations, $\varepsilon =10^{-3}$
led to deterioration in accuracy due to excessively coarsened resolutions on the non-uniform grid that were much coarser
than that of the DEM, and is therefore not recommended. For the urban-scale simulations at DEM resolutions as fine as 2 m
or lower, both $\varepsilon$ values were found valid for use with the non-uniform solver to preserve accuracy, but $\varepsilon =10^{-3}$ leads to more
sensible resolution coarsening.

       In terms of speed-ups, the GPU version of the uniform grid solver was increasing faster to run than the CPU version
as the number of elements on the grid became increasingly larger, reaching up to 28 times with 6 million elements. On the
GPU, the non-uniform solver could offer further enhancements in speed-up, depending on the DEM resolution and the

properties of the case study. For coarse DEM resolutions featuring the catchment-scale simulations at the recommended $\varepsilon$ $=10^{-4}$, the speed-up enhancement increased with more reduction in the number of elements, reaching up to 4 times for a reduction of 36 %. A similar enhancement was identified at finer DEM resolution featuring the urban-scale simulations at the recommended $\varepsilon =10^{-3}$, albeit at a higher reduction, exceeding 65 %, when the grid was dominated by at most two

resolutions. The enhancement became moderate to insignificant when there was more depth and breadth in resolutions with even distributions, which is likely the case where the GPU version of the uniform solver should be preferred. One future direction to add more efficiency for very large-scale simulations would be to locally integrate the subgrid channel model within the non-uniform solver on the GPU to further coarsen the portions including the wider river channels.

**Code and Data availability**

LISFLOOD-FP8.1 source code (LISFLOOD-FP developers, 2022; https://doi.org/10.5281/zenodo.6912932) and simulation results (Sharifian et al., 2022; https://doi.org/10.5281/zenodo.6907286) are available from Zenodo. Due to access restrictions, readers are invited to contact the Environment Agency for access to the data used in Sect. 3.4 and to refer to the cited references for the data used in Sects. 3.1, 3.2, 3.3 and 3.5.

**Video supplement**

Step by step instructions on how to download and install the LISFLOOD-FP8.1, and reproduce the simulations for representative case studies (Sects. 3.2 and 3.4) are available from Zenodo at https://doi.org/10.5281/zenodo.6685125 (Sharifian and Kesserwani, 2022).

**Author contributions**

MKS coded the numerical solvers in collaboration with AAC and JN. MKS and GK were responsible for the
conceptualization, methodology, formal analysis, investigation, visualization and writing the initial draft. All authors contributed to paper review and editing.

**Competing interests**

The contact author has declared that neither they nor their co-authors have any competing interests.

**Publisher's note**

Copernicus Publications remains neutral with regard to jurisdictional claims in published maps and institutional affiliations.

**Acknowledgements**

We wish to thank Ilhan Özgen-Xian (Technische Universität Braunschweig) for sharing the data for Lower Triangle catchment case study, and Xilin Xia (Loughborough University) for sharing the data for Upper Lee and Eden catchment case studies. This work is part of the SEAMLESS-WAVE project (SoftwarE infrAstructure for Multi-purpose fLood modElling at

various scaleS based on WAVElets, https://www.seamlesswave.com).

**Financial support**

Mohammad Kazem Sharifian, Georges Kesserwani and Alovya Ahmed Chowdhury were supported by the UK Engineering and Physical Sciences Research Council (EPSRC) grant EP/R007349/1. Jeff Neal was funded by the Natural Environment





Research Council (NERC) grant NE/S006079/1. Paul Bates was supported by a Royal Society Wolfson Research Merit
award.

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
