# Peer review of "LISFLOOD-FP 8.1: New GPU accelerated solvers for faster fluvial/pluvial flood simulations"

_Geoscientific Model Development, 2022_

## Author Comment (AC1)

Dear Referee #1,

We thank you for your time to review our paper and for the well-considered comments, which have helped us improve and better scope the discussion and conclusions sections of the paper.

Below, we will repeat each comment (italic font) and reply directly below it (standard front). After each reply, we flag the associated changes applied in the revised paper to ease the re-review.

Best wishes,
Georges Kesserwani and Mohammad Kazem Sharifian
* * *
*The manuscript titled "LISFLOOD-FP 8.1: New GPU accelerated solvers for faster fluvial/pluvial flood simulations" deals with the upgrade if the well-known LISFLOOD hydrodynamic simulator, using parallel programming and specifically the GPU capabilities in order to speed up the simulations. Except of the parallelization, the authors demonstrate the use of a smart grid coarsening way, which also speeds up the simulations but with an accuracy sacrify. The paper is well written and well structured and characterized by novelties.* Referee #1, appreciatively, recognised the aim, scope and novelties of this contribution, suggesting well-considered technical corrections that have been addressed as described below.

*I would suggest to be published after some minor technical corrections:*

*1) It is not consistent to compare all the numerical results (uniform, non-uniform 10^-3, non-uniform 10^-4) against the observed data. Since the non-uniform is an simplification of the uniform detailed grid, the latter should be the base of comparison and the observed values should be given as a supplementary material, not substantial for the core of the paper.* We agree with Referee #1 about this comment. Already, in the original manuscript, the quantitative results used the uniform detailed grid as the base. For consistency, the qualitative results have been revised to refer to the uniform detailed grid as the base too. As for the observed values, we have kept them along with the results of the uniform grid as a useful indication of the validity of the base model.

*The situation in which the non-uniform grid performs better than the uniform grid is rather a coincidence. I assume that the non-uniform grids introduce a kind of artificial diffusion, while similar results could be derived by the uniform grid with bigger values of Manning coefficients.* We have also elaborately discussed that the fact the non-uniform grids introduce artificial diffusion and cited a paper that confirms the assumption of Referee #1. The associated revised text can be seen in the box below, in the discussions of the discharge hydrographs in Figure 11:

> The predicted flow discharges at the outlet (post-processed hydrographs), are compared against the observed hydrograph in Figure 11. None of the simulated hydrographs closely trail the observed hydrograph, including that predicted by the uniform solver. This deficiency can be attributed to uncertainties in the location of the in-situ measurements, in the aggregation of the
>
> 335 simulated discharges at coarse resolutions that further magnify by larger numerical diffusions accumulating on the coarser portions of the static non-uniform grid (Kesserwani and Sharifian, 2023), in the inability of the present Manning's friction law formula to model rain-driven overland flows in the catchment areas with low Reynolds numbers (Kistetter et al., 2016; Taccone et al., 2020), and to the fact that the features of river channel bathymetry are not captured in the 20 m DEM. As shown in Kesserwani and Sharifian (2023), such issues could be alleviated to some extent by using dynamic grid adaptivity that deploys a
>
> 340 more complex formulation combining the multiresolution analysis (MRA) of the multiwavelets with a second-order discontinuous Galerkin (MWDG2) solver. However, capturing such events more accurately with the present solvers using static grid adaptivity and the ACC solver formulation seems to suggest a need for deploying a DEM resolution that is at most 10 m (Ferraro et al.,
>
> 12

And also in the discussions of the water level time series in Figure 21, as shown in the box below:

[revised manuscript text omitted]

---

## Author Comment (AC2)

Dear Ilhan,

Thank you for providing an all-round review, which has helped us clarify many aspects of relevance to flood model developers and practicioners.

Below, we will repeat each comment (italic font) and reply directly below it (standard front). After each reply, we flag the associated changes applied in the revised paper to ease the re-review.

Best wishes,
Georges Kesserwani and Mohammad Kazem Sharifian
* * *
*The authors present a GPU-accelerated shallow flow solver that is being incorporated into the well-known LISFLOOD framework. This effort to improve on existing and established software used by many practicioners is a timely and relevant endeavour. The improvements presented in the manuscript are (i) GPU-acceleration with specific focus on non-uniform Cartesian grids and (ii) wavelet-based mesh adaptation to generate these grids. The manuscript is well written and easy to follow. The selected test cases are meaningful and support the authors' claims.* We thank the reviewer for highlighting the relevance and timeliness of our contribution and value his all-around comments that have been addressed in the revised paper as described below.

*Therefore, I suggest accepting the manuscript after minor revisions.*

*1. Governing equations. 1.1 The authors omit showing the governing equations. At least for me, it made the introduction of this paper difficult to follow, specifically the discussion of the ACC solver and its differences to the fully dynamic shallow flow solver (page 1, lines 35—45). I would suggest showing the SWE explicitly, naming the acceleration term and momentum terms in these equations, and then showing which terms drop out in the ACC solver. 1.2 The acronym ACC is being used without explanation. Please provide the full name of this solver the first time you use the acronym.* We agree and have revised the introduction section to show the SWE explicitly with the proper naming of terms in these equations and how then simply to what is referred to as "ACC solver", which its acronym is introduced as soon as it appeared. Associated changes applied in the revised introduction section is given in the box below:

LISFLOOD-FP includes a variety of numerical schemes to solve the two-dimensional (2D) shallow water equations, which can be written as:

$$\frac{\partial h}{\partial t} + \frac{\partial q_x}{\partial x} + \frac{\partial q_y}{\partial y} = r \tag{1}$$

$$\underbrace{\frac{\partial q_x}{\partial t}}_{Acceleration} + \underbrace{\frac{\partial (gh^2/2)}{\partial x}}_{Pressure} + \underbrace{\frac{\partial (q_x^2/h)}{\partial x} + \frac{\partial (q_x q_y/h)}{\partial y}}_{Advection} + \underbrace{gh\frac{\partial z}{\partial x}}_{\substack{Bed \\ gradient}} + \underbrace{\frac{gn_M^2|q_x|q_x}{h^{7/3}}}_{Friction} = 0 \tag{2}$$

$$\underbrace{\frac{\partial q_y}{\partial t}}_{Acceleration} + \underbrace{\frac{\partial (gh^2/2)}{\partial x}}_{Pressure} + \underbrace{\frac{\partial (q_y^2/h)}{\partial y} + \frac{\partial (q_x q_y/h)}{\partial x}}_{Advection} + \underbrace{gh\frac{\partial z}{\partial y}}_{\substack{Bed \\ gradient}} + \underbrace{\frac{gn_M^2|q_y|q_y}{h^{7/3}}}_{Friction} = 0 \tag{3}$$

where Eq. (1) is the mass conservation equation and Eqs. (2-3) are the momentum conservation equations. In these equations $h$ [L] represents the water depth, and $q_x = hu$ and $q_y = hv$ [L²/T] are the discharge per unit width in the $x$ and $y$ orthogonal directions, respectively, expressed involving the velocity components $u$ and $v$ [L/T]; $r$ [L/T] is the prescribed rainfall rate, $g$ [L/T²] is the gravitational acceleration, $z$ [L] is the two-dimensional topographic elevation and $n_M$ is Manning's coefficient [L^{1/6}]. Along with the finite volume and discontinuous Galerkin schemes that solve the full form of Eqs. (1)-(3) (Shaw et al., 2021), two simplified schemes are also available in LISFLOOD-FP: a diffusive wave scheme, that neglects both acceleration and advection terms, and a local inertial scheme, that neglects the advection term but includes the acceleration term (hence, referred to as the ACC solver). The extra complexity of including the acceleration term in the ACC solver pays off with larger and more stable time steps than the diffusive wave solver (Hunter et al., 2006; Hunter et al., 2008; Bates et al., 2010; Wang et al., 2011). Compared to the finite volume and Galerkin schemes, the ACC solver is faster to run since it also does not involve an approximate Riemann solver (Shaw et al, 2021): it uses a hybrid finite difference/volume numerical scheme, in a staggered approach that evaluates continuous discharges across the grid elements to then achieve an element-wise update of averaged water levels (Bates et al., 2010). This

*2. Morton codes. 2.1 Out of curiosity, what upper bound does the use of Morton codes give you for the allowable number of cells? I am asking because, if I understood correctly, if you combine the binary representation of two integers into one, you can only use half the length that a computer can store for each representation. So on a 64 bit machine, this would mean that the maximum number of elements that can be used is what can be represented with 32 bits. Is this correct?* This is correct: since a Morton code is represented by a 64-bit signed integer, the two binary representations that are interleaved to produce the Morton code can use up to 32 bits each. Therefore, each binary representation is effectively represented by a 32-bit signed integer. A 32-bit signed integer can represent a maximum value of $2^{31}$ - 1 = 2147483647 (see https://en.wikipedia.org/wiki/2,147,483,647#In_computing). To clarify this, a short explanation has been added to the revised manuscript as shown in the box below:

To efficiently perform MRA on the GPU, a data structure based on the Z-order curve is deployed, which maps the hierarchy of grids to a single 1D array. A Z-order curve can be created for a $2^l \times 2^l$ grid by following the so-called Morton codes of the grid elements, which are obtained by bit interleaving the x and y indices of each element. Since a Morton code is represented by a 64-bit integer, the two binary representations that are interleaved to produce the Morton code can use up to 32 bits each. Therefore, each binary representation is effectively represented by a 32-bit integer. An example of the creation of a Z-order curve using the Morton codes for a $2^2 \times 2^2$ grid is shown in Figure 3. As seen in Figure 3a, X and Y indices are stored in binary form and bit

*3. Lower Triangle catchment. 3.1 Source of data should be completed with the source of the raw data: Wainwright, H. and Kenneth, W. (2017). LiDAR collection in August 2015 over the East River Watershed, Colorado, USA. https://doi.org/10.21952/WTR/1412542.* The relevant reference has been added as shown in the box below:

230    The case study reproduces a real-world rainfall-runoff event over a mountainous catchment considering three DEM resolutions of 2, 5 and 10 m, which were downscaled from the available 0.5 m DEM resolution in Wainwright and Williams (2017). It aims to assess the accuracy of the non-uniform grid solver with increasingly coarser DEM resolution. The catchment is a sub-catchment of the East River watershed in Colorado, USA, with a surface area of 14.84 km$^2$ featured by high-elevation topography, with steep slopes towards the main river stream located in the middle of the catchment (Figure 5). The 72-hour flood event was induced by a

235    storm that occurred in August 2016, based on the rainfall time series shown in Figure 5, which was given at 30-minute intervals (Wainwright and Williams, 2017). The friction parameter is uniformly set to $n_M = 0.035$ m$^{1/6}$ over the domain that was initially dry before a simulation starts. A simulation starts by loading the temporally-varying rainfall data, uniformly into all elements on the grid.

*4. Fluvial vs. pluvial test cases. Perhaps some of the answers to the comments below could be placed in the Conclusions and recommendations section.* In addition to revising the Conclusions and recommendations section to address the comments of Referee #1, this section has also been revised in light of the answers provided to the following comments.

*4.1 From the test cases and my own experience, mesh coarsening seems to "work better" for fluvial runoff, probably because pluvial runoff yields very small water depths that elevate the influence of the topography. This is to some extent supported by the authors' results. Can the authors comment?* As correctly pointed out by the reviewer, the small water depths that often arise in the pluvial floods makes the flow more sensitive to the shape of the terrain. This effect is even more significant when considering the ACC solver, where the shallow water flows over steep slopes with high velocities, emerging as supercritical flows. The resulting instabilities could adversely affect the performance of the ACC solver both in terms of accuracy and efficiency. Based on our results, the non-uniform grid might smooth those steep slopes to some extent, lowering the chance of forming supercritical flows. The discussions of the Lower Triangle case study have been revised to clarify this, as shown in the box below:

    The accuracy of the non-uniform solver predictions for the flood extent is analysed in Figure 7, which shows the maps of

255    the difference in maximum flood extents for the two $\varepsilon$ values. With $\varepsilon = 10^{-4}$, at 2 m DEM resolution, the predicted maximum flooded area shows the least agreement with the extent predicted by the uniform grid solver, with a C of 0.56, leading to pronounced underpredictions (red areas, H = 0.69) and overpredictions (blue areas, F = 0.24) over the tributary streams (Figure 7a). The relatively poor scores are related to the inherent limitation of the ACC solver in modeling thin fast-flowing, water depths occurring over a very steep terrain from the high intensity runoff (De Almeida et al 2012; Sridharan et al., 2021). This is also

260    noted for the coarser 5 and 10 m DEM resolutions, where the overall agreement of the predictions by the non-uniform grid solver becomes close to the uniform grid counterparts, leading to higher C values of 0.61 and 0.67, respectively. Note that, improved scores are expected with the mesh coarsening (as discussed in Figure 6a) since it reduces the steepness of the topographic slopes,

9

which in turn reduces the occurrence of thin and high-velocity flows. The same tendency is observed for the larger $\varepsilon = 10^{-3}$, with a more notable loss of accuracy in maximum flood extent predictions that occur due to the higher reductions in the number of

265    elements on the non-uniform grid (Figure 6b). This loss of accuracy reduces the C values to 0.43, 0.44 and 0.48 for 2, 5 and 10 m DEM resolutions, respectively (Figure 7b). However, the C values remain close to each other at $\varepsilon = 10^{-3}$, while remaining lower than 0.5, suggesting that this $\varepsilon$ value may be too large for catchment-scale simulations over DEMs at a resolution of 2 m or lower and should, therefore, be avoided (also shown next in Sects. 3.2 and 3.3).

Also, our results confirm the reviewer's comment, that shallow water depths are less probable to happen in case of fluvial floods.

*4.2 The hydrograph of the Upper Lee catchment shows that coarser grids damp short time-scale events. This has been my experience with multiresolution meshes as well. Are there mitigations the authors suggest that could lead to more accurately capturing these short time-scale events?* We agree with the reviewer and have discussed this issue further by elaborating on the impact of numerical diffusion, due to local mesh-coarsening (as suggested by Referee #1) and, suggested one solver option (i.e., the multiwavelet based second order discontinuous Galerkin solver, a.k.a, MWDG2), which has been recently proven to preserve the accuracy of the uniform solver on multiresolution meshes, by being more resistant to the accumulation/growth of numerical diffusion and offering a more sensible resolution coarsening. Nevertheless, we have avoided the use of the term "short time-scale" for this event as it might not be the case for a 120 hours long flooding scenario. The associated changes applied in the revised paper are shown below:

> The predicted flow discharges at the outlet (post-processed hydrographs), are compared against the observed hydrograph in Figure 11. None of the simulated hydrographs closely trail the observed hydrograph, including that predicted by the uniform solver. This deficiency can be attributed to uncertainties in the location of the in-situ measurements, in the aggregation of the
> 335 simulated discharges at coarse resolutions that further magnify by larger numerical diffusions accumulating on the coarser portions of the static non-uniform grid (Kesserwani and Sharifian, 2023), in the inability of the present Manning's friction law formula to model rain-driven overland flows in the catchment areas with low Reynolds numbers (Kistetter et al., 2016; Taccone et al., 2020), and to the fact that the features of river channel bathymetry are not captured in the 20 m DEM. As shown in Kesserwani and Sharifian (2023), such issues could be alleviated to some extent by using dynamic grid adaptivity that deploys a
> 340 more complex formulation combining the multiresolution analysis (MRA) of the multiwavelets with a second-order discontinuous Galerkin (MWDG2) solver. However, capturing such events more accurately with the present solvers using static grid adaptivity and the ACC solver formulation seems to suggest a need for deploying a DEM resolution that is at most 10 m (Ferraro et al.,
>
> 12
>
> 2020), which is not available for this case study. Therefore, the predictability of the solvers to the flow discharge at the outlet may

*4.3 The pluvial flooding in the Glasgow urban area is very well captured, compared to Lower Triangle and Lee catchments. Is this due to the regularity of the urban area?* As correctly pointed out by the reviewer, the better performance of the models for the Glasgow case study is deemed to be related to the regularity and smoothness of the terrain, being situated in a low-lying area, which (as explained above) lowers the chance of supercritical flows to happen. This is further clarified in the revised discussion of the test case, as shown in the box below:

> 485    This accuracy of the non-uniform solver predictions at $\varepsilon = 10^{-3}$ can be confirmed by comparing the results in Figure 18, which shows its predicted time series of the water levels to be comparable with those predicted the uniform solver, all agreeing with the reference predictions. At all the gauge points, the non-uniform solver, with both $\varepsilon = 10^{-3}$ and $10^{-4}$, leads to almost similar accuracy compared to uniform solver predictions, successfully reproducing the double peak observed within the reference predictions and other predictions reported in the literature (Neelz and Pender, 2013). Hence, the non-uniform solver, with both $\varepsilon =$
> 490    $10^{-3}$ and $10^{-4}$, is a valid alternative to the uniform solver for such a case study featured by flooded zones within a small domain area with smooth terrain that is, moreover, represented at a fine DEM resolution of 2 m.